



# Prediction of natural dry-snow avalanche activity using physics-based snowpack simulations

Stephanie Mayer[1,*], Frank Techel[1,*], Jürg Schweizer[1], and Alec van Herwijnen[1]

[1]WSL Institute for Snow and Avalanche Research SLF, Davos, Switzerland
[*]joint first authorship

**Correspondence:** Stephanie Mayer (stephanie.mayer@slf.ch)

**Abstract.** Accurately predicting the location, timing and size of natural snow avalanches is crucial for local and regional decision-makers, but remains one of the major challenges in avalanche forecasting. So far, forecasts are generally made by human experts, interpreting a variety of data, and drawing on their knowledge and experience. Using avalanche data from the Swiss Alps and one-dimensional physics-based snowpack simulations, we developed a model predicting the probability

of dry-snow avalanches occurring in the vicinity of automated weather stations based on the output of a recently developed instability model. This new avalanche day predictor was compared to benchmark models related to the amount of new snow. Evaluation on an independent data set demonstrated the importance of snow stratigraphy for natural avalanche release, as the avalanche day predictor outperformed the benchmark model based on the three-day sum of new snow (F1 scores: 0.71 and 0.65, respectively). The averaged predictions of both models resulted in the best performance (F1 score: 0.75). In a second

step, we derived functions describing the probability for certain avalanche size classes. Using the 24-hour new snow height as proxy of avalanche failure depth yielded the best estimator of typical (median) observed avalanche size, while the depth of the deepest weak layer, detected using the instability model, provided the better indicator regarding the largest observed avalanche size. Validation of the avalanche size estimator on an independent data set of avalanche observations confirmed these findings. Furthermore, comparing the predictions of the avalanche day predictors and avalanche size estimators with a 21-year data set of

re-analysed regional avalanche danger levels showed increasing probabilities for natural avalanches and increasing avalanche size with increasing danger level. We conclude that these models may be valuable tools to support forecasting the occurrence of natural dry-snow avalanches.

## 1 Introduction

Forecasting natural snow avalanches is highly relevant in areas where avalanches may threaten people or infrastructure. Erro-

neous forecasts may cause costs as (1) missed alarms may result in damage to people or infrastructure, and (2) false alarms may lead to economic loss due to unnecessary closures or evacuations. Therefore, accurately predicting the occurrence of natural avalanches in space and time is crucial, though still a major challenge in avalanche forecasting. Currently, forecasts are made by human experts, drawing on their knowledge and experience. To forecast natural dry-snow avalanches, the (expected) amount of new snow is one of the main parameters. Accumulated sums of precipitation were found to be among the most important



explanatory variables in several studies relating observed avalanche activity to meteorological drivers and observed snowpack parameters (e.g. Ancey et al., 2004; Kronholm et al., 2006; Hendrikx et al., 2014). However, new snow depth alone is not sufficient for forecasting, but other contributing factors, in particular the presence of potential weak layers in the snowpack, have to be taken into account (e.g. Stoffel et al., 1998; Schirmer et al., 2009; Schweizer et al., 2009).

While physical snowpack models, such as CROCUS (Brun et al., 1989, 1992; Vionnet et al., 2012) or SNOWPACK (Lehn-
ing et al., 1999; Bartelt and Lehning, 2002; Lehning et al., 2002a, b), are commonly used to model new snow amounts for operational avalanche forecasting, they have so far only rarely been used to assess snowpack instability based on simulated snow stratigraphy in an operational context (Morin et al., 2020). Some recent studies included information on simulated snow stratigraphy as explanatory variables to predict natural avalanche activity with statistical or machine learning models (Viallon-Galinier et al., 2022; Reuter et al., 2022). Viallon-Galinier et al. (2022) found a Random Forest (RF) classifier that included
mechanically-based stability indices to outperform a classifier that only relied on meteorological and bulk snow parameters simulated with CROCUS. However, the precision of the improved classifier was low (3.4%), which was attributed to the scarcity of avalanche events and the potential misclassification of non-avalanche days in the observations. The uncertainty inherent in avalanche observation data generally poses a major challenge when developing avalanche prediction models. Errors in visual observations arise from the difficulty of retrospectively determining the exact date of an avalanche release and from
missed avalanche events due to limited visibility during periods of heavy snowfall, when the probability of natural avalanche events is particularly high. Avalanche activity data recorded by automatic detection systems (e.g. Heck et al., 2019; Mayer et al., 2020) is a promising alternative, but commonly covers only very limited areas (a few $\text{km}^2$), much smaller than typical forecasting regions (order of $100 \text{ km}^2$). Moreover, due to the relatively new technologies of automatic avalanche detection, avalanche catalogues only cover a few winter seasons (van Herwijnen et al., 2016). For instance, Reuter et al. (2022) trained
and tested a model using automatically detected avalanches using only 31 non-avalanche days and 15 avalanche days.

An alternative approach to develop snow instability models is to use a target variable based on surrogate data that implicitly contain information on avalanche activity, e.g. avalanche danger levels or stability test results from field observations. According to the definitions of the European avalanche danger levels (EAWS, 2021), natural avalanches are expected at level 4 (high) and 5 (very high), but unlikely at the two lowest levels (1 (low), 2 (moderate)). In addition, avalanche size increases with
increasing danger level (e.g. EAWS, 2022; Schweizer et al., 2020a; Techel et al., 2020). Pérez-Guillén et al. (2022) recently developed a RF classifier that uses meteorological parameters and snow-cover properties simulated with SNOWPACK to predict danger levels. Another recent RF classifier was trained on stability tests related to human-triggered avalanches (Mayer et al., 2022). This model, which we refer to as the *instability model* in the following, assesses the probability that a simulated SNOWPACK profile is potentially unstable considering human triggering. As the instability model was trained using stability
tests related to human-triggered avalanches, its applicability to predict natural avalanches is not self-evident. However, its input features describing the potential weak layer (e.g. grain size) and the overlying slab (e.g. the ratio of the mean slab density and the mean slab grain size) are important variables not only with respect to human triggering but also regarding natural release. Comparing the classification of SNOWPACK profiles simulated using measurements from more than 100 automated weather stations (AWS) in Switzerland with a large number of avalanche forecasts, showed plausible results: the instability model





yielded low probabilities of instability if the forecast danger level was low (i.e. level 1 (low) or 2 (moderate)) or in aspects and at elevations not indicated as critical in the forecast; whereas high probabilities were predicted for the upper danger levels (i.e. level 3 (considerable) or 4 (high)) (Techel et al., 2022). The instability model was tested in an operational setting by the national avalanche warning service in Switzerland during the 2021/2022 winter season, with promising results.

The objective of this study is to investigate whether the instability model developed by Mayer et al. (2022) applied to one-
dimensional SNOWPACK simulations can be used to predict natural dry-snow avalanches. More specifically, we aim to derive a transformation of the current model output (probability of instability) to an index describing the probability of observing natural dry-snow avalanches in the surrounding of an AWS. For this purpose, we use avalanche observations recorded for avalanche forecasting in Switzerland during three winter seasons and SNOWPACK simulations from automated weather stations located at the elevations of potential avalanche starting zones. To reduce the uncertainty associated with visual avalanche observations,
we apply a filter using observations from the wider surroundings. Furthermore, as a secondary objective, we explore whether we can estimate avalanche size based on one-dimensional SNOWPACK simulations. The avalanche day predictor and the avalanche size estimator are both validated using 21 years of re-analysed regional danger level data and an independent data set of avalanche observations (5 years) from the region of Davos in the eastern Swiss Alps. With these validation data, we also demonstrate the usefulness of predictions based on the instability model compared to the use of simple indicators of snow
instability as the amount of new snow during the previous 24 or 72 hours.

## 2 Data

We used different data sets to train and validate the avalanche day predictor and the avalanche size estimator (Fig. 1). To develop the avalanche day predictor, we used avalanche observations (data set AV1, Sect. 2.1.1) combined with SNOWPACK simulations and predictions of the instability model described in Section 2.2. The avalanche size estimator was trained using
only avalanche observations (data set AV2, Sect. 2.1.2). For validation of both models, we used a third independent data set of avalanche observations (data set AV3; 2.1.3), as well as a data set of quality-checked regional avalanche danger levels (DL; Sect. 2.3).

### 2.1 Avalanche data

### 2.1.1 Swiss Alps, observed avalanches (data set AV1, 2019/2020 - 2021/2022, 3 years)

To develop the avalanche day predictor, and test the avalanche size estimator, we used avalanche observations collected for the purpose of avalanche forecasting in Switzerland. During the winter season, generally from early December until late April, about 80 observers report avalanches in their region on a daily basis. These observations are highly relevant for the day-to-day verification of the avalanche forecast, particularly at the higher danger levels. Reported avalanche properties include the approximate location and the date of the avalanche release, the elevation and the slope aspect of the release area, the release
type (i.e. natural or human-triggered), whether it was a dry- or a wet-snow avalanche (SLF, 2020), and a size estimate according



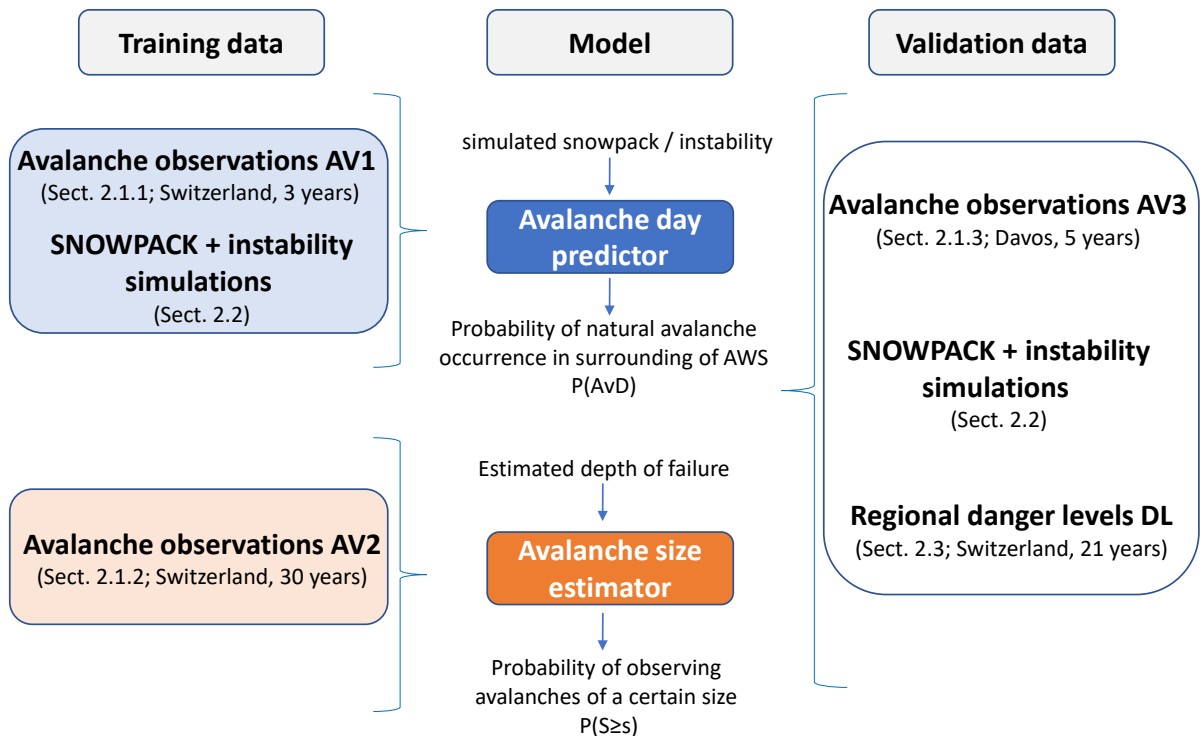

**Figure 1.** Several data sets were used to develop and validate the functions describing the probability of natural avalanche occurrence and avalanche size. The data are described in the sections indicated.

to the European avalanche size classification ranging from 1 (small) to 5 (extremely large) (EAWS, 2021, see Table 1). In many cases, the release date and time but also other parameters are estimated, as the actual avalanche release was not observed and access to the starting zone of an avalanche is generally not possible. Other avalanche characteristics like the type of avalanche (i.e. slab or loose snow avalanche), the length and width, or the failure depth are also reported sometimes.

For this study, we only considered natural dry-snow avalanches that were recorded between 1 December and 30 April in the three winter seasons 2019-2020, 2020-2021, and 2021-2022 in the Swiss Alps. In total, 12'940 avalanches were reported (Tab. 3). Even though the operational avalanche database also contains avalanche observations prior to 2019, the recording standards were different and did not allow us to unambiguously identify natural dry-snow avalanches.

### 2.1.2   Swiss Alps, observed avalanches (data set AV2, 1992/1993 - 2021/2022, 30 years)

The data set described in Sect. 2.1.1 only rarely contained an estimate of avalanche failure depth, which is equal to slab thickness. To derive a relationship between the failure depth of avalanches and avalanche size, we therefore extracted all dry-snow avalanches that contained an estimation of avalanche size and the (mean estimated) failure depth from the operational database. Between November 1992 and June 2022 (30 years), this resulted in 5912 dry-snow avalanches.



**Table 1.** Avalanche size classification (according to SLF, 2020; EAWS, 2021) and the corresponding weight, $w$, used for the calculation of the $AAI$.

| size ($s$) | label | volume [m$^3$] | weight ($w$) |
|---|---|---|---|
| 1 | small | 100 | 0.01 |
| 2 | medium | 1'000 | 0.1 |
| 3 | large | 10'000 | 1 |
| 4 | very large | 100'000 | 10 |
| 5 | extremely large | >100'000 | 10 |

### 2.1.3 Davos/Eastern Swiss Alps, observed avalanches (2014/2015 - 2018/2019, 5 years, data set AV3)

For validation, we used avalanches mapped in the region of Davos in the eastern Swiss Alps (e.g described by Hafner et al., 2021). These data were used in several studies (e.g. Schweizer et al., 2020a; Mayer et al., 2022), and are publicly available (Schweizer et al., 2020b). From an updated version of this data set, we extracted all natural dry-snow avalanches that released in the 5 winters 2014/2015 to 2018/2019.

### 2.2 Snowpack and instability simulations

We applied the operational setup of the SNOWPACK model (Lehning et al., 2002b) used for avalanche forecasting in Switzerland. The simulations were driven with meteorological data from automatic weather stations (AWS) located in flat terrain at the elevation of potential avalanche starting zones throughout the Swiss Alps (Lehning et al., 1999; Morin et al., 2020). An overview of the spatial distribution of these AWSs is provided in Figure 2. The measured meteorological data were pre-processed with MeteoIO (Bavay and Egger, 2014) to filter out potential measurement errors and fill measurement gaps using

temporal interpolation or spatial interpolation with neighbouring stations. To reduce errors related to the meteorological input data in the validation (4.4.1) of the models developed in this study, we also produced SNOWPACK simulations using a quality-checked data set of meteorological measurements from the AWS Weissfluhjoch (2536 m a.s.l.) (WSL Institute for Snow and Avalanche Research SLF, 2015).

In addition to the simulations on flat terrain, simulations were also performed for four 'virtual' slope orientations (N, E, S,

W) with a slope angle of 38°. Model output was available for up to 124 AWSs. We used SNOWPACK simulations from the 21 winters 2001/2002 until 2021/2022.

To assess snow instability from simulated snow stratigraphy, we applied the instability model to the simulated snow profile at 12:00 LT on the day of interest, as described in Techel et al. (2022). The instability model requires six input features describing the simulated snow layer of interest and the overlying slab. The output probability $P_{\text{unstable}}$ that a snow layer is unstable is

determined by the fraction of trees in the ensemble of 400 classification trees that classify the layer as potentially unstable.





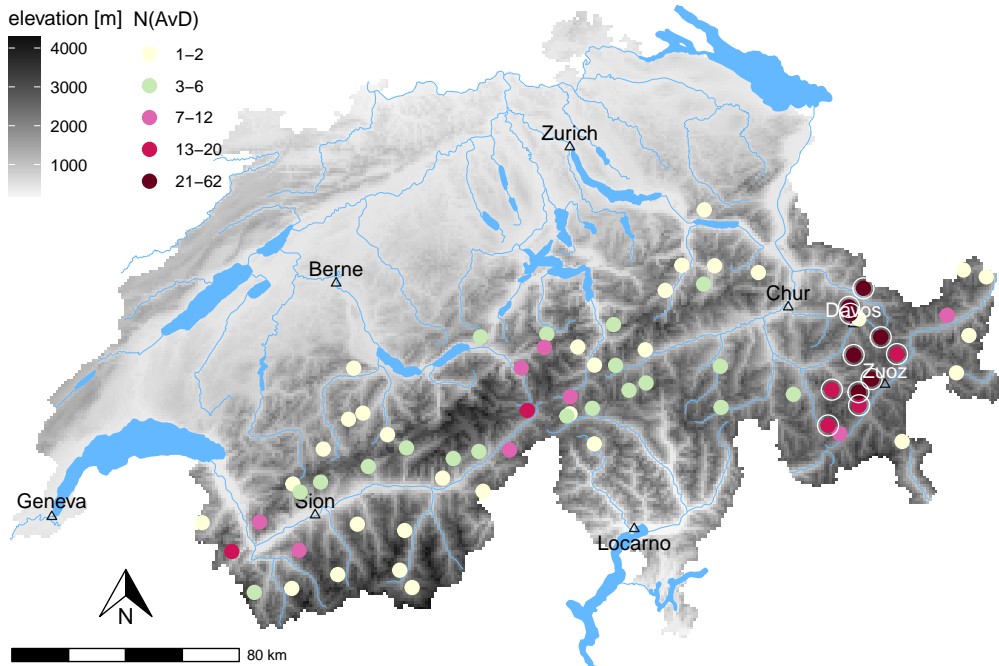

**Figure 2.** Map of Switzerland showing the location of the automated weather stations (dots). The coloring indicates the number of avalanche days (AvD) per station. Stations in the Davos-Zuoz area, which had $N \geq 13$ AvDs per station were combined in a subset «Davos-Zuoz» (marked with white circles), all other stations as «elsewhere». For illustration purposes, the major rivers and lakes are shown in blue, and the elevation in grey. (Digital elevation model - source: Federal Office of Topography swisstopo)

Applying the instability model to every snow layer of a given snow profile allows computing the following properties (see also Figure 3):

- **Critical weak layer properties**: The **critical weak layer** relevant for natural avalanche release is defined as the layer with the highest probability of instability, i.e. the layer where $P_{\mathrm{unstable}} = \max(P_{\mathrm{unstable}})$. In case of ties, we selected the layer deepest in the snowpack. For each snow profile, we then extracted the following three layer properties: $\max(P_{\mathrm{unstable}})$, which we refer to as $P_{\mathrm{crit}}$, the depth $z$ below the snow surface in cm ($z_{\mathrm{crit}}$), and the grain type ($gt_{\mathrm{crit}}$). We grouped grain types into three classes considering the primary grain type: (i) persistent grain types ($pg$), including depth hoar, buried surface hoar, facets, and rounding facets, (ii) precipitation particles ($pp$), including decomposing and fragmented precipitation particles, and (iii) other grain types ($other$), including rounded grain types, melt forms, melt-freeze crusts and ice layers (see also Fierz et al. (2009) for the grain type classification).

- **Deepest weak layer properties**: In addition to the critical weak layer, we searched for potential weak layers deeper in the snowpack. We selected the **deepest weak layer** as the deepest layer fulfilling $P_{\mathrm{unstable}} \geq 0.77$ - the best-splitting threshold suggested by Mayer et al. (2022) to distinguish between stable and potentially unstable layers. If no such



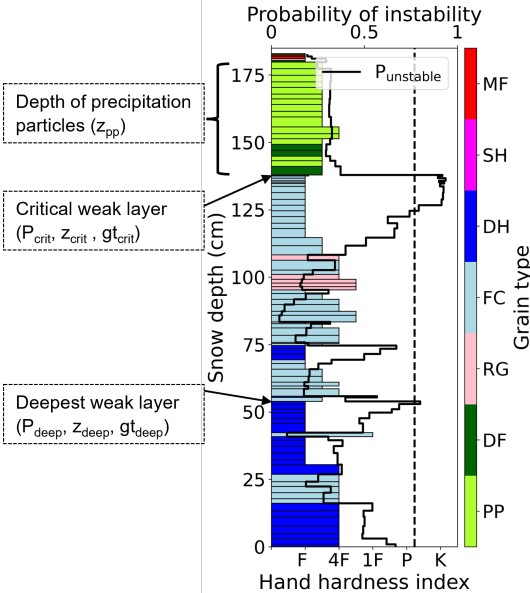

**Figure 3.** Example of a simulated snow profile showing the hand hardness profile, the grain type of the simulated layers (coloring of the layers), and the probability of instability $P_{\text{unstable}}$ (black line). The critical weak layer is defined as the layer where $P_{\text{unstable}}$ is maximal. Hand hardness (F - fist, 4F - four fingers, 1F - one finger, P - pencil, and K - knife) and grain type (PP - precipitation particles, DF - decomposing and fragmented precipitation particles, RG - rounded grains, FC - faceted crystals, DH - depth hoar, SH - surface hoar, MF - melt forms) were coded after Fierz et al. (2009). The dashed vertical line displays the threshold of $P_{\text{unstable}} = 0.77$ discriminating between stable and potentially unstable layers derived by Mayer et al. (2022). The depth of precipitation particles and the deepest weak layer, i.e. the deepest layer exceeding the instability threshold, are indicated.

layer existed, the deepest weak layer was the critical weak layer. For each profile, we then extracted the probability of
instability for the deepest weak layer $P_{\text{deep}}$, the depth below the snow surface ($z_{\text{deep}}$), and the grain type ($gt_{\text{deep}}$).

The rate of snowfall and the amount of new snow are known to be important indicators of natural dry-snow avalanche activity, also called direction-action avalanches (e.g. Conway and Wilbour, 1999), but also for the potential size of avalanches (e.g. Schweizer et al., 2009). Therefore, we also calculated:

– **Height of new snow in 24 hours** (hn1d).

– **Height of the three-day sum of new snow** (hn3d), calculated as the sum of three consecutive hn1d-values .

– **Depth of precipitation particles**: The thickness of layers in the simulated profile, where the primary grain type was either new snow or partially decomposing and fragemented precipitation particles ($z_{\text{pp}}$).

And lastly, we also extracted the minimum of the natural stability index sn38, implemented in SNOWPACK (Lehning et al., 2004).




Of these parameters, the two parameters describing the height of new snow were independent of aspect, while all other parameters were extracted from the simulated profile, and thus vary between the four virtual slope aspects. For instance, energy input due to radiation varies dependent on aspect. In addition, the operational SNOWPACK version includes snow redistribution as described in Lehning et al. (2000) and Lehning and Fierz (2008).

## 2.3    Re-analysed regional avalanche danger level (data set DL)

To validate model predictions, we used a data set of re-analysed regional danger levels. This data set is a subset of the forecast regional avalanche danger levels published by the national avalanche warning service in Switzerland. The data set only contains cases for which the forecast danger level was either validated or corrected (about 5% of the cases) after considering multiple pieces of evidence, as described by Pérez-Guillén et al. (2022). An updated version of this data set is publicly available (Techel, 2022). The data set consists of 36'582 re-analysed regional danger levels for specified warning regions, the smallest spatial

units used in the Swiss avalanche forecast, for the forecast seasons from winter 2001/2002 to 2021/2022. In addition, the critical aspects and elevation range where the danger level applies, and the validity date of the forecast are indicated. The danger level is assigned according to the five-level European Avalanche Danger Scale (EAWS, 2022). The frequency of the danger levels in this data set is: 1 (low) 35%, 2 (moderate) 29%, 3 (considerable) 29%, 4 (high) 7%, 5 (very high) 0.3%. In this re-analysed subset, the proportions for 4 (high) and 5 (very high) are slightly larger than in the original forecasts.

## 3    Methods

In a first step, we developed an *avalanche day predictor* (Sect. 3.1) addressing the question: for a given $P_{\mathrm{crit}}$-value, what is the probability of natural avalanches occurring in a specific aspect and elevation band in the surroundings of an AWS. We compared this approach with benchmark models based on conventional indicators related to the amount of new snow. Second, we built an *avalanche size estimator* (Sect. 3.2) with the objective to extract information on the expected typical or largest

avalanche size based on (simulated) weak layer depth or the height of new snow.

### 3.1    Avalanche day predictor

#### 3.1.1    Definition of avalanche days and non-avalanche days

To discriminate days with natural dry-snow avalanche activity (avalanche days, *AvD*s) from days without any avalanche activity (non-avalanche days, *nAvD*s), we relied on data set AV1 (Sect. 2.1.1). The two main challenges in using these data relate to

reliably labeling days with no avalanches and the correct estimation of the release date. For instance, even in areas that are regularly observed, the absence of reported avalanches may be due to poor visibility (i.e. continuous snowfall) rather than a true absence of recent avalanches, making it challenging to accurately determine situations without natural avalanches. Moreover, the accuracy of the release date depends on observation frequency in an area, on visibility conditions and the overall observation



quality. To enhance the reliability of the avalanche day labels, we therefore applied the approach developed by Hendrick et al.
(accepted) to extract AvDs and nAvDs from the avalanche observations, with a specific focus on dry-snow avalanches.

We define the avalanche day index ($Y$) in the vicinity of an automated weather station $st$ and within an elevation band
$\pm 250$ m of the station elevation for four slope orientations (aspect $asp =$: N, E, S, W) as:

$$Y = \begin{cases} 0 & \text{if } AAI(SwissAlps) = 0 \text{ for all elevations and aspects} \\ 1 & \text{if } AAI(250 \text{ km}^2) \geq 0.01 \ \& \ AAI(1000 \text{ km}^2) \geq 0.04 \ \& \\ & \quad AAI(5000 \text{ km}^2) \geq 0.2 \ \& \text{ gap check} \\ \text{NaN} & \text{otherwise}, \end{cases} \quad (1)$$

with NaN not a number. The avalanche activity index $AAI$ refers to the weighted sum of the reported avalanches within the
respective elevation band, aspect and area (Schweizer et al., 2003). The size-dependent weights $w$ are defined as in Table 1.
The gap check requirement is $AAI(5000 \text{ km}^2) > AAI(1000 \text{ km}^2) \ \& \ AAI(1000 \text{ km}^2) > AAI(250 \text{ km}^2)$, ensuring avalanche
activity increases for larger areas and is not only local. As described in Hendrick et al. (accepted), considering observation
areas of different size allows to cross-check the absence or occurrence of avalanches.

This definition separates days with widespread avalanche activity (AvD; $Y = 1$) of a certain magnitude from days with
absolutely no avalanches (nAvD; $Y = 0$), and excludes days with either only local avalanche activity (close to the station) or
widespread activity but without any avalanches in the vicinity of the station. Regarding model development, it should be noted
that we thus trained and tested our model using rather extreme cases, which are, however, comparably reliable in terms of the
quality of the label.

After classifying data set AV1 into AvDs and nAvDs, which in the following we will refer to as the training data set, the data
set contained about ten times more nAvDs ($N = 8511$) than AvDs ($N = 872$). Overall, AvDs had a median of two avalanches
(interquartile range IQR: 2-7) in the aspect and elevation of the snowpack simulation within an area of 250 km$^2$ surrounding
the station. Two or more avalanches were recorded on 559 of the 872 AvDs. The median AAI was 1 (IQR 0.3-3.6). The typical
avalanche (median avalanche size) was of size 2 (IQR 2-3) and the largest of size 3 (IQR 2-3).

### 3.1.2 Model development and evaluation

To develop the avalanche day model, we tested a set of predictor variables including hn1d, hn3d, $z_{pp}$ and $P_{crit}$ in two different
modelling approaches, namely a threshold-based binary classification model and continuous regression functions describing
the probability for an AvD.

In a first step, we investigated the performance of each predictor variable in discriminating between AvDs and nAvDs from
the training data set using a simple threshold-based binary classification model. To find the best threshold for each classification
model, we optimized the F1 score, defined as the harmonic mean of the precision, also termed positive predictive value (PPV),
and the true-positive rate (TPR; see Table 2 for the definitions of these performance measures). This approach favors a balanced
trade-off between the TPR, which is the probability of detecting an AvD, and the PPV, the rate of correct positive predictions.





**Table 2.** Confusion matrix defining the possible combinations of observed and predicted labels (upper part) and definition of resulting performance measures true-positive rate (TPR), positive predictive value (PPV), true-negative rate (TNR) and F1 score (F1; lower part).

|  |  | Observation | |
|---|---|---|---|
|  |  | 1 (AvD) | 0 (nAvD) |
| Prediction | 1 | TP | FP |
|  | 0 | FN | TN |
| TPR | = | $\frac{TP}{TP+FN}$ | |
| PPV | = | $\frac{TP}{TP+FP}$ | |
| TNR | = | $\frac{TN}{TN+FP}$ | |
| F1 | = | $2\frac{PPV \cdot TPR}{PPV+TPR}$ | |

To examine the robustness of the threshold values and the resulting classification performance, we split the data AV1 into several subsets, each of which was tested with the complementary data not used for deriving $thr$. We split by:

– Hydrological year: We split the data by hydrological year, each with its own pattern of snowpack evolution and avalanche hazard characteristics.

    – Grain type characteristics of critical weak layer: We distinguished between layers composed of persistent grain types and precipitation particles. There were only a few AvD cases for other grain types, therefore we did not train on this subset.

    – Region: The AvDs are not equally distributed over the Swiss Alps (see Fig. 2). Ten of the 11 stations with the most

AvDs are all located in the eastern Swiss Alps, in an area we refer to as *Davos-Zuoz*. This region is characterized by an inner-alpine climate. To ascertain that the threshold was independent of this spatial bias in the data, we compared a subset Davos-Zuoz (black dots in Figure 2) to *elsewhere*.

In a second step, we derived avalanche day predictors $P(\text{AvD})$ describing the probability for an AvD as continuous functions of a single input feature, i.e.

$$P(\text{AvD})(x) = f(x), \quad \text{with } x = \text{hn1d}, \text{hn3d}, z_{\text{pp}} \text{ or } P_{\text{crit}}. \tag{2}$$

To estimate the relationship between the binary avalanche index data $Y \in \{0, 1\}$ and the predictor variables we applied regression analysis with four-parameter sigmoidal (S-shaped) functions (see Table A1 in the appendix). The functions were fit on the complete training data set using non-linear least squares with parameter constraints to ensure that modeled probabilities did



not exceed 1. For each input feature, we defined the best fitting function by minimizing the Brier score (*BS*; Wilks, 2011, p. 331), which is the mean squared prediction error:

$$\text{BS} = \frac{1}{N} \sum_i (Y_i - p_i)^2, \tag{3}$$

where $N$ is the number of the prediction-observation samples denoted with index $i$, $p_i$ the predicted probability - here $f(x_i)$, and $Y_i$ the observed outcome (1 for AvD and 0 for nAvD). To indicate how well the minority class of AvD was captured by the model, we additionally calculated the Brier score on the subset of AvDs only (BS$^+$).

## 3.2 Avalanche size estimator

To estimate avalanche size for a given failure layer depth, we used data set AV2 (Sect. 2.1.2) to relate avalanche size to observed failure depth ($z_\text{obs}$) using logistic regression functions of the form

$$P(S \geq s)(z_\text{obs}) = \frac{1}{1 + e^{-(\beta_0 + \beta_1 \cdot z_\text{obs})}}, \tag{4}$$

where $P(S \geq s)$ is the probability that avalanches greater or equal than size $s$ ($s \in [2, 3, 4, 5]$; Table 1) were observed given the observed failure depth ($z_\text{obs}$).

The $P(S \geq s)$-functions were derived using observed data only, as SNOWPACK simulations were not available for the locations of the avalanche release areas. To analyze the performance of the size indicators combined with the depth parameters extracted from SNOWPACK, we estimated probabilities for different avalanche sizes on the AvDs from the training data set (Section 3.1.1) using the simulated depth parameters $z$ (hn1d, hn3d, $z_\text{pp}$, $z_\text{crit}$, $z_\text{deep}$, described in Section 2.2) as proxies for the potential failure depth. The resulting estimated probabilities for different avalanche sizes were then compared to the observed median and maximum avalanche sizes on the respective AvD using the Brier score (eq. 3) with the probabilities $p_i = P(S \geq s)(z_i)$, and the observed outcome $Y_i$ equal to 1 if an avalanche of size $\geq s$ was observed and 0 otherwise. We also calculated the Brier score on the subset of positive observed outcomes (BS$^+$) to evaluate how the avalanche size estimator performed when predicting potentially rare events.

## 3.3 Validation and application

To evaluate the performance of the avalanche day predictors ($P(\text{AvD})$) and the avalanche size estimators ($P(S \geq s)$), we used two independent data sets:

1. We used the observations of natural dry-snow avalanches in the region of Davos (data set AV3, Sect. 2.1.3) to determine AvDs and nAvDs as described in Section 3.1.1. Due to the lack of information on trigger type, liquid water content, aspect and elevation of the observed avalanches in the surrounding regions, we somewhat adapted the definitions. We labeled a day as a nAvD if there were no dry-snow avalanches in the region of Davos and the neighbouring regions





(1000 km$^2$ and 5000 km$^2$). We labeled a day as an AvD on a specific aspect, if at least one natural dry-snow avalanche was observed within each of the two neighbouring regions (1000 km$^2$ and 5000 km$^2$), regardless of aspect and elevation. The resulting data set consisted of 273 avalanche days and 984 non-avalanche days during the five winter seasons 2014-2015 until 2018-2019. For each of these 1257 days, we calculated aspect-specific values of $P(\mathrm{AvD})$ and $P(S \geq s)$ using SNOWPACK virtual slope simulations driven with quality-checked data from the AWS Weissfluhjoch (see Sect. 2.2).

2. We compared the re-analysed forecast regional avalanche danger levels (DL, Sect. 2.3) to $P(\mathrm{AvD})$- and $P(S \geq s)$ values computed for the stations and virtual slopes that matched the elevation and the critical aspects of the respective danger level data point. For the winter seasons 2019-2020 to 2021-2022, we removed all data points used to develop the $P(\mathrm{AvD})$-model, and which had a simulated snow depth $< 30$ cm.

## 4 Results

### 4.1 Avalanche days vs. non-avalanche days

Avalanche days were generally associated with new snow in the last 24 hours (hn1d $= 25$ cm, hn3d $= 59$ cm, $p < 0.001$, row $= all$ in Tab. 3). In contrast, nAvDs were typically characterized by no new snow (hn1d $= 0$, hn3d $= 0$, median values, $p < 0.001$). Consequently, the median slab depth consisting of precipitation particles varied in a similar way (AvD : $z_{\mathrm{pp}} = 73$ cm, nAvD : $z_{\mathrm{pp}} = 0$ cm). The simulated critical weak layer was at a median depth of 75 cm on AvDs and 22 cm on nAvDs. The simulated critical weak layer had a significantly higher probability of instability on AvDs compared to nAvDs ($P_{\mathrm{crit}} = 0.92$ vs. $P_{\mathrm{crit}} = 0.33$, respectively, $p \leq 0.001$) and it was more often composed of persistent grain types (77% vs. 47% of cases, respectively, $p \leq 0.001$). As indicated in Table 3, these values varied between subsets. For instance, the median depth of the most critical weak layer was 44 cm on AvDs in 2020 and 91 cm in 2021, while on nAvDs the values were 4 cm and 65 cm, respectively. Similarly, on AvDs, the depth of the weak layer was 88 cm when the critical weak layer consisted of persistent grains (pg) compared to 47 cm for precipitation particles (pp).

At least one potentially unstable layer was detected in 84% of the AvDs, and in only 2% of the nAvDs. Moreover, in 7% of the profiles, there was at least one other potentially unstable layer below the critical weak layer. These cases were rare on nAvDs (1% of the profiles), but quite frequent on AvDs (66%). The median difference in the depth between the critical and the deepest potentially unstable layer ($z_{\mathrm{deep}} - z_{\mathrm{crit}}$) was 14 cm (IQR: 4-44 cm). On AvDs, these layers were 15 cm deeper (IQR: 5-49 cm) compared to only 4 cm (IQR: 2-26 cm) on nAvDs. If such a deeper weak layer existed, it primarily consisted of persistent grains (90%).

### 4.2 Predicting avalanche days and non-avalanche days

All explored variables (hn1d, hn3d, $z_{pp}$ and $P_{\mathrm{crit}}$) showed highly significant differences between avalanche days and non-avalanche days as demonstrated in the previous section. In the following, we will first explore their potential for a binary classification of AvDs and nAvDs, and then derive continuous functions describing the probability for an AvD.



The optimal thresholds ($thr$) to distinguish between nAvD and AvD for the seven subsets varied when cross-validating the model. For instance, threshold values ranged from 9 to 17 cm for the 24-hour amount of new snow hn1d (median 13 cm) or from 22 to 47 cm for $z_{pp}$ (median 32 cm) (Appendix, Table A2). Applying these thresholds to the test sets, i.e. the data not used for training, showed that all four variables performed similarly well in correctly predicting nAvDs (TNR $\in [0.96, 1]$; Fig. 4). In contrast, larger variations were observed in the true positive rate TPR, that is the proportion of correctly predicted AvDs. TPR was highest for hn3d (TPR $= 0.81$) and $P_{crit}$ (TPR $= 0.79$). The precision, i.e. the proportion of predicted AvDs that also were observed as AvDs, was highest for the two new snow parameters (PPV(hn1d) $= 0.84$, PPV(hn3d) $= 0.83$). However, these two parameters also showed a greater variation in PPV between subsets compared to $P_{crit}$, which had a more consistent performance though a slightly lower PPV of 0.80. Overall, in terms of a balanced performance maximizing the F1 score, both hn3d and $P_{crit}$ had similar values (median F1 score of 0.80 in cross-validation). All approaches by far outperformed the natural stability index sn38 (median cross-validated F1 score of 0.24).

Analyzing differences between the subsets in more detail also provided interesting insights. For instance, the optimal balanced $z_{pp}$-threshold to differentiate AvD from nAvD was 40 cm when the critical weak layer consisted of precipitation particles (*pp*) compared to 22 cm for persistent grains *pg*; it was 47 cm in the region *elsewhere* and 22 cm in the inner-alpine region of *Davos-Zuoz*, where persistent weak layers are more frequently observed (e.g. Schweizer et al., 2021, see also Table 3). Similar results were also obtained for the two new snow variables, thus confirming what is known from a process-based point of view: when persistent weak layers are present, less new snow is needed to trigger natural avalanches (Stoffel et al., 1998; Schweizer et al., 2009).

The coefficients for the best-fitting sigmoidal functions $f$ yielding the avalanche day estimators $P(\text{AvD})(x) = f(x)$ with x given by hn1d, hn3d, $z_{pp}$, or $P_{crit}$ are shown in the Appendix Table A3. The Brier Score BS was lowest for hn3d (BS $= 0.021$, $BS^+ = 0.156$; Table A3). Exemplary, Figure 5 shows the $P(\text{AvD})$-functions for hn3d and $P_{crit}$. The values of $P(\text{AvD})$ predicted with these two variables correlated strongly (Pearson correlation coefficient $r = 0.82$). The thresholds where the functions reached $P(\text{AvD}) = 0.5$ (Table A3) were slightly higher compared to the thresholds of the binary classification models described above (Table A2). The F1 scores resulting from these thresholds deviated from the optimal F1 scores obtained with the simple classifiers by less than 1%. Therefore, we only evaluated the performance of the continuous avalanche day predictor functions in the validation (Sect. 4.4).

Finally, we explored the performance when averaging the $P(\text{AvD})$-predictions based on hn3d and $P_{crit}$. Taking the mean of both models resulted in slightly better performance compared to the best performing approach $P(\text{AvD})(hn3d)$: the Brier Score BS decreased from 0.021 to 0.019, while the Brier Score for predictions of positive events $BS^+$ decreased from 0.156 to 0.144. Translating the mean probability into a binary classification resulted in a TPR of 0.81, a TNR of 0.99, and a high PPV of 0.95. Thus, the combined model detected more than 80% of the avalanche days correctly, and had the overall highest F1 score of 0.87.




**Table 3.** Overview of avalanche data and properties of simulated most critical weak layer as selected by the instability model and thickness of overlying slab consisting of precipitation particles (recent slab). Subsets are shown by hydrological year (2020, 2021, 2022), by region (Davos-Zuoz area and elsewhere, see also Fig. 2), and as function of the grain type of the critical weak layer (pg = persistent grain types, pp = precipitation particles). The seven numbered subsets were used for cross-validation. Median values are shown for hn1d, hn3d, $P_{crit}$, $z_{crit}$ and $z_{pp}$, and the proportion of critical weak layers consisting of persistent grain types ($gt_{crit} = pg$).

| | avalanche activity | | | | new snow | | | | recent slab | | properties critical weak layer | | | | | | | |
| subset | N(aval) | AAI(CH) | N(AvD) | N(nAvD) | hn1d (cm) | | hn3d (cm) | | $z_{pp}$ (cm) | | $P_{crit}$ | | $z_{crit}$ (cm) | | $gt_{crit} = pg$ | |
| | | | | | AvD | nAvD | AvD | nAvD | AvD | nAvD | AvD | nAvD | AvD | nAvD | AvD | nAvD |
| (1) 2020 | 3733 | 1262 | 138 | 1911 | 16 | 0 | 31 | 0 | 37 | 0 | 0.83 | 0.30 | 44 | 4 | 0.75 | 0.34 |
| (2) 2021 | 6231 | 4941 | 542 | 3440 | 31 | 0 | 68 | 0 | 85 | 0 | 0.92 | 0.38 | 91 | 65 | 0.75 | 0.48 |
| (3) 2022 | 2976 | 1196 | 192 | 3160 | 18 | 0 | 44 | 0 | 50 | 0 | 0.93 | 0.32 | 59 | 16 | 0.76 | 0.53 |
| (4) Davos-Zuoz | – | – | 557 | 4175 | 22 | 0 | 53 | 0 | 65 | 0 | 0.92 | 0.34 | 72 | 18 | 0.80 | 0.51 |
| (5) elsewhere | – | – | 315 | 4336 | 30 | 0 | 68 | 0 | 86 | 0 | 0.92 | 0.33 | 83 | 21 | 0.67 | 0.43 |
| (6) pg | – | – | 655 | 3973 | 23 | 0 | 59 | 0 | 70 | 0 | 0.93 | 0.41 | 88 | 70 | – | – |
| (7) pp | – | – | 205 | 563 | 29 | 2 | 60 | 6 | 82 | 7 | 0.89 | 0.48 | 47 | 4 | – | – |
| (-) other | – | – | 12 | 3970 | 14 | 0 | 21 | 0 | 20 | 0 | 0.39 | 0.28 | 32 | 3 | – | – |
| all | 12940 | 7399 | 872 | 8511 | 25 | 0 | 59 | 0 | 73 | 0 | 0.92 | 0.33 | 75 | 22 | 0.77 | 0.47 |



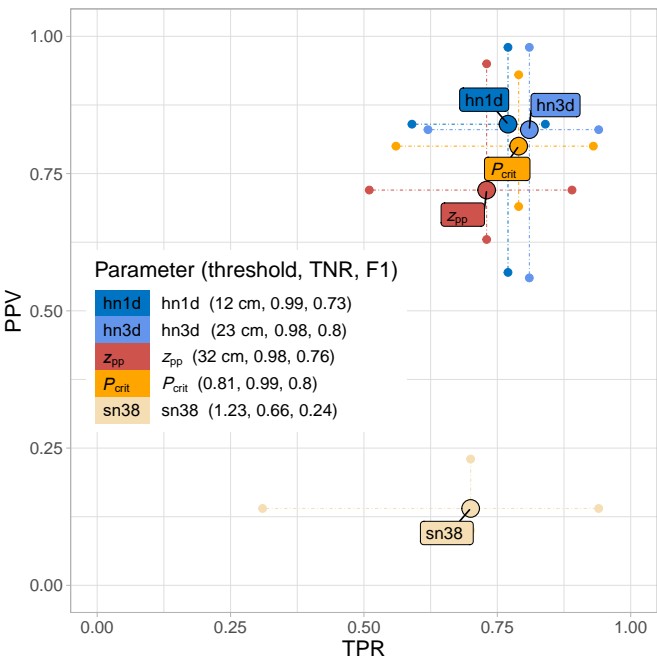

**Figure 4.** Performance statistics for avalanche day predictors (binary classification). Shown are the cross-validated TPR- and PPV-values for the seven subsets numbered in Table 3. The whiskers mark the respective minimum and maximum values, the larger circles display the median values of these performance measures on the seven subsets. In addition, for each parameter, the median threshold, TNR, and F1 score are indicated in the legend.

## 4.3 Estimating avalanche size

In data set AV3, containing 5912 observed avalanches (Sect. 2.1.2), the recorded failure depth $z_{obs}$ correlated with avalanche size ($r_s = 0.45$, $p < 0.001$; Fig. 6a). The median failure depth increased from 30 cm for size 1 avalanches to 100 cm for size 5 avalanches. While there is considerable overlap, the distributions of $z_{obs}$ were significantly different between pairs of consecutive avalanche size classes (Wilcoxon rank-sum test $p < 0.001$). Based on this data set, we derived logistic functions $P(S \geq s)(z_{obs})$ to estimate avalanche size from $z_{obs}$ (Figure 6b; the respective coefficients are provided in the Appendix in Table A4).

Comparing $P(S \geq 3)$ and $P(S \geq 4)$ with the observations from the data set AV1 on AvDs with at least two recorded avalanches, we obtained the lowest Brier score BS if the median avalanche size was estimated with hn1d as a proxy for the failure depth. For the largest recorded avalanche, on the other hand, $z_{deep}$ was the best predictor (Table 4). Considering only subsets when the avalanche size of interest was indeed observed ($BS^+$ in Table 4), i.e. for the 175 cases $S \geq 3$ when an avalanche of size 3 or larger was observed, $z_{deep}$ had the lowest Brier scores for both the median and the largest avalanche size. Thus, $z_{deep}$ outperforms the variables related to the amount of new snow in terms of capturing positive minority events even for the median avalanche size, but has a tendency to predict avalanches larger than observed.





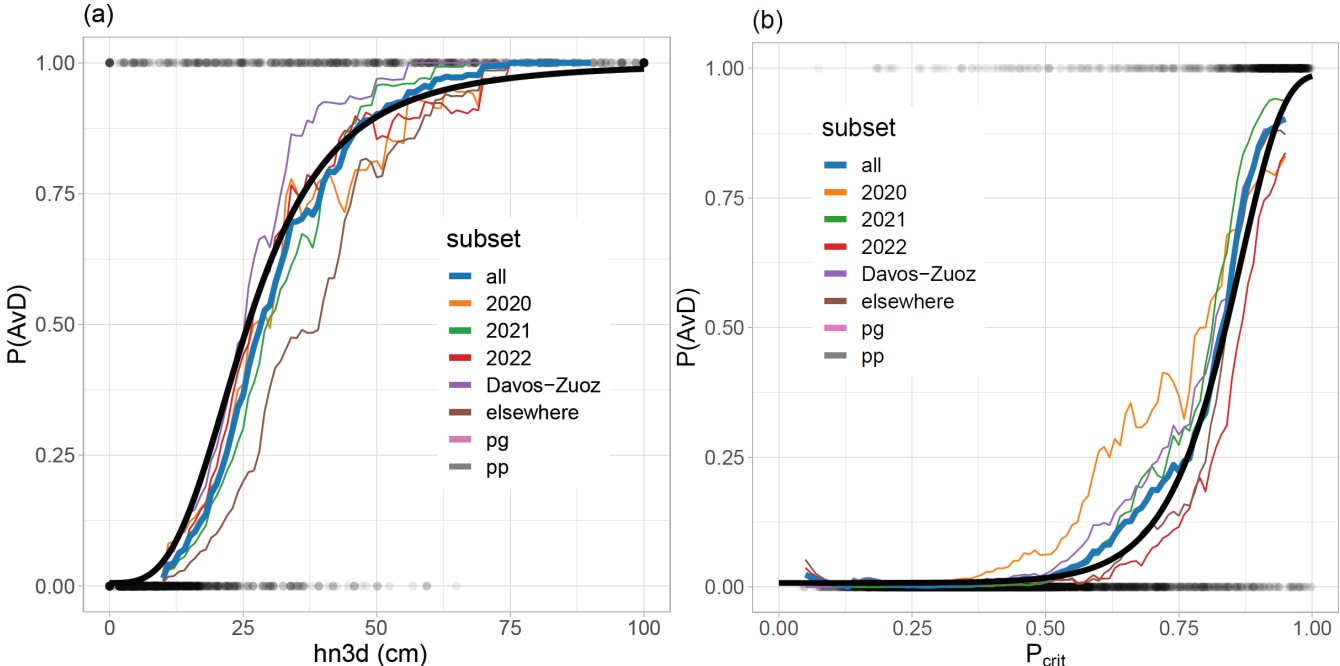

**Figure 5.** Probability that a day is an avalanche day ($P(\mathrm{AvD})$) as a function of (a) hn3d and (b) $P_{\mathrm{crit}}$ for the data subsets shown in Table 3. The subsets are binned with bin-size being 10 cm in (a) and 0.1 in (b). The best-fitting function describing all data is shown in black.

**Table 4.** Brier scores BS for predicting the median or the largest avalanche for all avalanche days (AvD) with $\geq 2$ avalanches ($N = 559$) using as input different predictor variables. BS$^+$ evaluates only a subset of the data when the condition is fulfilled (median / maximum $S \geq 3$: N = 175 / N = 436; median / maximum $S \geq 4$: N = 31 / N = 140). The best-performing approach is highlighted bold.

|  |  | median avalanche size | | | | | largest avalanche size | | | | |
|---|---|---|---|---|---|---|---|---|---|---|---|
|  | $S \geq s$ | hn1d | hn3d | $z_{\mathrm{crit}}$ | $z_{\mathrm{deep}}$ | $z_{\mathrm{pp}}$ | hn1d | hn3d | $z_{\mathrm{crit}}$ | $z_{\mathrm{deep}}$ | $z_{\mathrm{pp}}$ |
| BS | $S \geq 3$ | **0.21** | 0.28 | 0.32 | 0.37 | 0.28 | 0.39 | 0.27 | 0.24 | **0.21** | 0.25 |
|  | $S \geq 4$ | **0.05** | 0.06 | 0.07 | 0.09 | 0.05 | 0.23 | 0.21 | 0.19 | **0.19** | 0.19 |
| BS$^+$ | $S \geq 3$ | 0.45 | 0.27 | 0.19 | **0.13** | 0.20 | 0.47 | 0.28 | 0.21 | **0.16** | 0.24 |
|  | $S \geq 4$ | 0.86 | 0.73 | 0.49 | **0.36** | 0.55 | 0.90 | 0.79 | 0.65 | **0.53** | 0.71 |

## 4.4 Validation

### 4.4.1 Predicting natural avalanche activity in the region of Davos

330

While the predictive power of the continuous models $P(\mathrm{AvD})(P_{\mathrm{crit}})$ and $P(\mathrm{AvD})(\mathrm{hn3d})$ was similar when applied to the training data set AV1 (see Table A3), there were substantial differences in the performance of these models on the validation





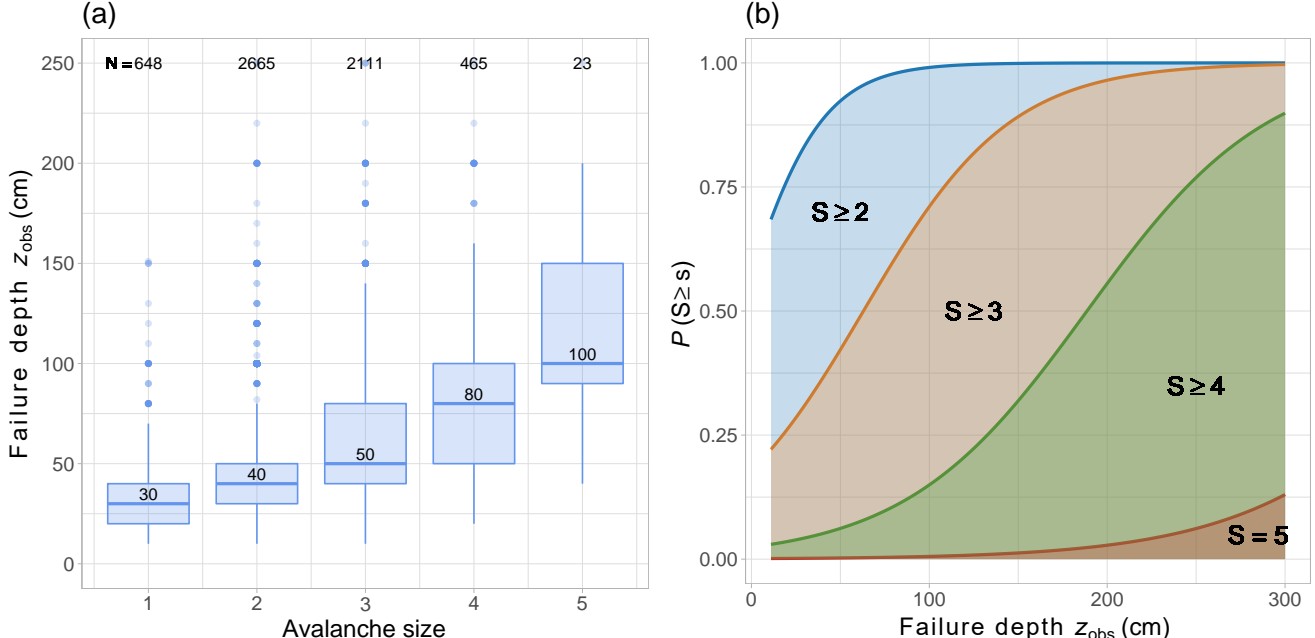

**Figure 6.** (a) Distribution of mean estimated failure depth ($z_{obs}$) for 5912 avalanches as a function of avalanche size. (b) Visualisation of logistic regression functions describing the probability that the avalanche size S is larger than a certain size s ($P(S \geq s)$) as a function of $z_{obs}$. The coefficients for these functions are shown in Appendix Table A4.

data set AV3 of observed avalanches from the region of Davos (Sect.2.1.3), as seen in Table 5 and Fig. 7. For both models, the predicted AvD-probability was low for nAvDs (median values 0.03 and 0.01, respectively), yet for AvDs, the $P(\text{AvD})$-

335  values based on hn3d were significantly lower (median: 0.55) than the values obtained with $P_{crit}$ (median: 0.95). Using the $P(\text{AvD})(\text{hn3d})$-model and the default classification threshold of 0.5, 149 of the 273 slope-specific AvDs were correctly predicted (TPR = 0.55) and 77% of the 193 predicted AvDs corresponded to actual observed AvDs (PPV = 0.77, Table 5). The $P(\text{AvD})(P_{crit})$-model, on the other hand, had a higher probability of detecting AvDs (TPR = 0.90), while the proportion of predicted AvDs that matched an observed AvD was lower (PPV = 0.59). In terms of F1 score, the $P(\text{AvD})(P_{crit})$-model

340  (F1 = 0.71) outperformed the $P(\text{AvD})(P_{\text{hn3d}})$-model (F1 = 0.64). An even higher F1 score of 75% was obtained when using the averaged probability of both models. This combined model yielded the highest precision (PPV = 0.79), but with a TPR of 0.72, less AvDs were detected than by the $P(\text{AvD})(P_{crit})$-model alone.

To evaluate the performance of the avalanche size estimators, we compared $P(S \geq 3)$-values estimated using hn1d or $z_{deep}$ with the observed median and maximum avalanche size on AvDs with at least two observed avalanches. The resulting Brier

345  scores shown in Table 6 are in line with the performance for data set AV1 (Table 4): The lowest Brier score BS for the estimation of median avalanche size was obtained when using hn1d, while for the largest observed avalanche $z_{deep}$ was again the better





**Table 5.** Performance statistics of different avalanche day predictors $P(\mathrm{AvD})$ on the independent validation data set (AV3) with observed avalanches from the region of Davos including 273 AvDs and 984 nAvDs.

| model | TPR | TNR | PPV | F1 |
|---|---|---|---|---|
| $P(\mathrm{AvD})(\mathrm{hn3d})$ | 0.55 | 0.96 | 0.77 | 0.64 |
| $P(\mathrm{AvD})(P_{\mathrm{crit}})$ | 0.90 | 0.83 | 0.59 | 0.71 |
| $P(\mathrm{AvD})(\mathrm{combi})$ | 0.72 | 0.95 | 0.79 | 0.75 |

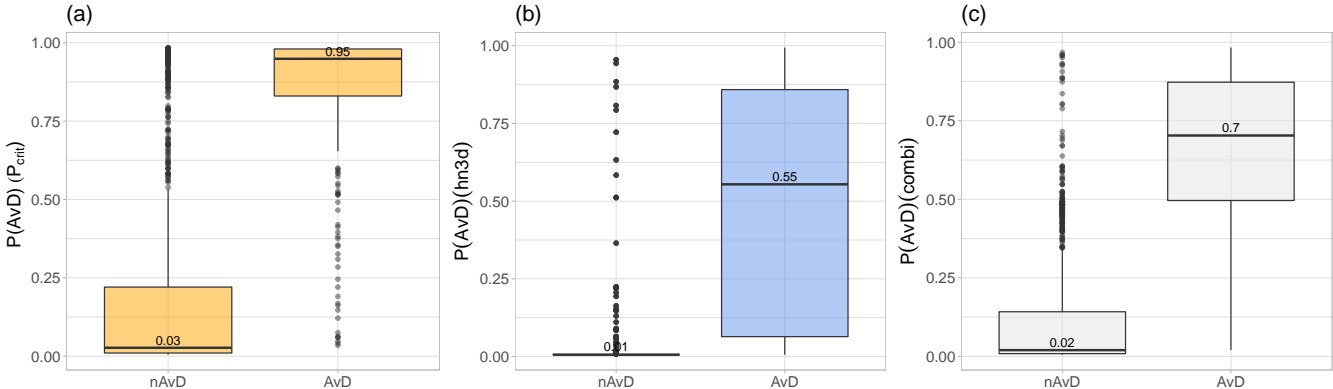

**Figure 7.** Estimated probabilities $P(\mathrm{AvD})$ for avalanche days (AvD) and non-avalanche days (nAvD) based on the data set of observed avalanches from the region of Davos using the models based on a) $P_{\mathrm{crit}}$, b) hn3d and c) the averaged predictions $P(\mathrm{AvD})(\mathrm{combi})$ of the models based on $P_{\mathrm{crit}}$ and hn3d.

**Table 6.** Brier scores BS for predicting the median or the largest avalanche size for all avalanche days (AvD) with $\geq 2$ avalanches ($N = 185$) from the validation data set AV3 of the region of Davos using the predictor variables hn1d and $z_{\mathrm{deep}}$ as input for the $P(S \geq 3)$-function. $\mathrm{BS}^+$ evaluates only a subset of the data when the condition is fulfilled (median / maximum $S \geq 3$: N = 33 / N = 126). The best-performing approach is highlighted bold.

| | | median avalanche size | | largest avalanche size | |
|---|---|---|---|---|---|
| | $S \geq s$ | hn1d | $z_{\mathrm{deep}}$ | hn1d | $z_{\mathrm{deep}}$ |
| BS | $S \geq 3$ | **0.15** | 0.39 | 0.39 | **0.23** |
| $\mathrm{BS}^+$ | $S \geq 3$ | 0.52 | **0.08** | 0.55 | **0.15** |

predictor. Considering only events when median avalanches sizes greater or equal than size 3 were observed (N= 33), again using $z_{\mathrm{deep}}$ resulted in the lower error rate $\mathrm{BS}^+$.





### 4.4.2 Comparison with the regional avalanche danger level

350 We compared individual model predictions with the quality-checked regional avalanche danger level for 21 winter seasons (data set DL; Sect.2.3). After removing data used for the development and testing of the $P(\text{AvD})$-models, 98 065 data points remained.

The proportion of simulated profiles which included a critical weak layer classified as potentially unstable $P_{\text{crit}} \geq 0.77$ increased significantly from danger level 1 (low) (0.01) to 2 (moderate) (0.19) to 3 (considerable) (0.61) (Figure 8a). At 355 the higher danger levels, the vast majority of the simulated critical weak layers were classified as potentially unstable (4 (high): 0.81, 5 (very high): 0.91). The median predicted probabilities for natural avalanches using the avalanche day predictor $P(\text{AvD})(P_{\text{crit}})$ were low at danger level 1 (low) (0.01) and 2 (moderate) (0.02), and increased with increasing danger level (3 (considerable): 0.56, 4 (high): 0.81, 5 (very high): 0.87) (Figure 8b). At the two lowest danger levels, less than 13% of the profiles indicated an AvD, while at the two highest danger levels more than 77% of the data points were classified as 360 AvD. The benchmark model $P(\text{AvD})(\text{hn3d})$ showed a similar increase in predicted avalanche probabilities with increasing danger level and differentiated even more clearly between the two lowest danger levels and the two highest danger levels (proportions $\leq 0.04$ and $\geq 0.82$, respectively) (Figure 8c). For both $P(\text{AvD})$-models, danger level 3 (considerable) had the largest spread in simulated avalanche probabilities. With the default threshold of 0.5, the proportion of data points at danger level 3 (considerable) that were classified as AvD by the $P(\text{AvD})(P_{\text{crit}})$- and $P(\text{AvD})(\text{hn3d})$-predictors were 54% and 43%, 365 respectively.

Avalanche sizes, estimated using the 24-hour new snow height hn1d, were mostly size 1 (proportions 0.34-0.42) and size 2 (proportions 0.4-0.44) for danger levels 1 (low) to 3 (considerable) (Figure 9a). At these danger levels, hn1d was 0 cm in 64% of the cases, and hence similar size distributions resulted. At 4 (high) and 5 (very high), new snow was recorded 94% of the time. The most frequently predicted avalanche size at the upper danger levels was size 2 (proportions 0.47-0.48), followed by 370 size 3 (0.26-0.38).

The second proxy of failure depth, $z_{\text{deep}}$ increased continuously with increasing danger level and obtained the following median values for 1 (low) to 5 (very high): 31 cm, 31 cm, 53 cm, 90 cm, and 154 cm. According to Mayer et al. (2022), the instability model detects the critical weak layer, and hence the depth of the weak layer, reliably only if the critical weak layer is potentially unstable. Taking this finding into account, and considering only $z_{\text{deep}}$ of weak layers rated as potentially unstable, 375 i.e. $P_{\text{deep}} >= 0.77$, the median depth of the weak layer, $z_{\text{deep}}$, led to similar median values of $z_{\text{deep}}$ ranging from 29 cm at 1 (low) to 157 cm at 5 (very high) (Figure 9e). Based on these values of $z_{\text{deep}}$, the most frequently predicted avalanche size was size 2 (proportions 0.45-0.48) at 1 (low) and 2 (moderate), and size 3 (proportions 0.43-0.48) at the three highest danger levels. While the proportions of size 3 were approximately similar at the three highest danger levels, the combined proportions of size 4 and size 5 avalanches increased considerably with increasing danger level from 0.13 at 3 (considerable) to 0.41 at 5 (very 380 high) (Figure 9b).



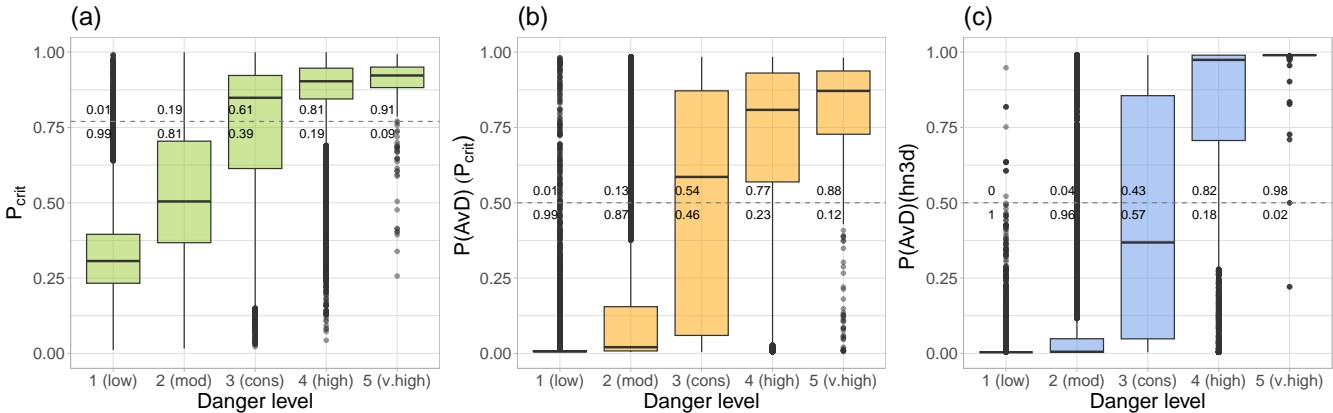

**Figure 8.** Comparison between quality-checked regional danger levels for 21 years (data set DL, N = 98 065, entire Swiss Alps; 1 (low), 2 (moderate), 3 (considerable), 4 (high), 5 (very high)) and simulated snow instability in terms of (a) the probability of instability of the critical weak layer, $P_{\mathrm{crit}}$, (b) the probability of an AvD provided by the avalanche day predictor $P(\mathrm{AvD})(P_{\mathrm{crit}})$, and (c) the probability of an AvD based on $P(\mathrm{AvD})(\mathrm{hn3d})$, the benchmark model. Model predictions were computed for the stations and virtual slopes that matched the elevation and the critical aspects of the respective danger level data point. The dashed horizontal line represents the best-splitting threshold to distinguish between (a) stable and potentially unstable profiles (0.77; Mayer et al., 2022), and (b, c) between AvDs and nAvDs. The respective proportions above and below this threshold are indicated for each danger level.

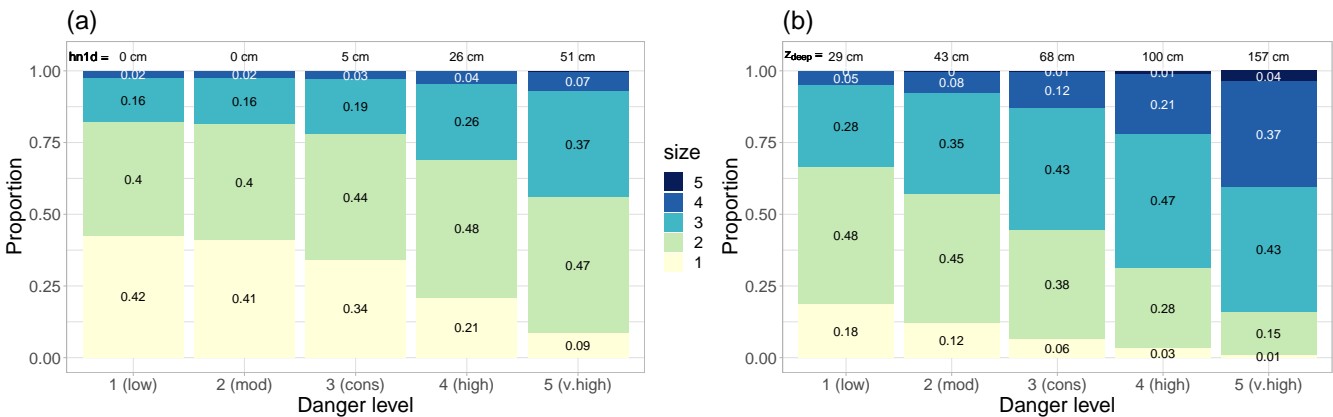

**Figure 9.** Comparison of quality-checked regional danger levels for 21 years (data set DL, N = 98 065, entire Swiss Alps; 1 (low), 2 (moderate), 3 (considerable), 4 (high), 5 (very high)) with simulated avalanche size distributions relying on the avalanche size estimators based on (a) hn1d and (b) $z_{\mathrm{deep}}$. For each danger level, the respective estimated proportions are shown for each avalanche size (colored bars). Median values of (a) hn1d and (b) $z_{\mathrm{deep}}$ are indicated at the top of the bars for each danger level.



## 5   Discussion

We developed an avalanche day predictor $P(\mathrm{AvD})(P_{\mathrm{crit}})$ describing the probability for natural dry-snow avalanches in the surrounding of an AWS for a given slope aspect based on simulated snow stratigraphy. We compared the performance of this index with benchmark models relying on the amount of new snow. The combination of $P(\mathrm{AvD})(P_{\mathrm{crit}})$ with a model based

on the 3d sum of new snow height, $P(\mathrm{AvD})(\mathrm{hn3d})$, yielded the overall best performance (Sect. 4.2 and 4.4.1). In a second step, we derived an avalanche size estimator based on the relationship between the reported failure depth of avalanches and avalanche size, providing the probability of observing avalanches of a certain size using different approximations of potential failure depth. The depth of the deepest weak layer, $z_{\mathrm{deep}}$, indicated by the instability model, was a better indicator of maximum avalanche size than modelled new snow amounts (Sect. 4.3 and 4.4.1). In the following, we will discuss the performance and

limitations of the avalanche day predictors ($P(\mathrm{AvD})$) (5.2) and avalanche size estimators ($P(S \geq s)$).

### 5.1   Data reliability

To develop the avalanche day predictor, we created a robust binary target variable (AvD versus nAvD) imposing restrictions on the observed avalanche activity in the vicinity of the AWS (Eq. 1), ensuring a high reliability of the labeling. With this approach, the target variable included rather extreme cases of widespread activity versus no activity at all, which should be

taken into account in model interpretation. As the exact timing of avalanche release was not included in the data sets of observed avalanches, the explanatory variables were extracted from the snowpack and instability model simulations at fixed time steps (12:00 LT). This introduced uncertainty in the explanatory variables of both the training and validation data sets. With avalanche data sets from remote detection systems, providing the exact release time, this uncertainty would be removed. However, so far such data sets only cover short time periods and are very local in scope (e.g van Herwijnen et al., 2016; Heck

et al., 2019; Mayer et al., 2020; Reuter et al., 2022), whereas the training data set used in this study included avalanches from the entire Swiss Alps observed over 3 winter seasons.

### 5.2   Predicting avalanche days

In a first step, we analyzed the predictive power of the explanatory variables to distinguish between AvDs and nAvDs using different subsets of the training data set (AV1). An optimized threshold-based classification resulted in a reasonably high

performance (cross-validated F1 score: 0.80) of $P_{\mathrm{crit}}$, and clearly outperformed the conventional natural stability index sn38 (cross-validated F1 score: 0.24). The poor performance of sn38 is in line with other studies (Schweizer et al., 2006; Jamieson et al., 2007). In contrast, the 3d new snow height (hn3d), recognized as an important indicator of avalanche activity in past studies (Ancey et al., 2004; Schweizer et al., 2009), yielded a classification performance (cross-validated F1 score: 0.80) similar to that obtained with $P_{\mathrm{crit}}$. Interestingly, the depth of precipitation particle layers $z_{\mathrm{pp}}$ resulted in a slightly lower classification

performance (cross-validated F1 score: 0.76), although it presumably captures the complete snowfall event, in contrast to hn3d. Potentially, using the mass of recent slab layers, which is more directly related to the load on the weak layer, may lead to better results than the depth of these layers.





Evaluating continuous one-dimensional sigmoidal $P(\text{AvD})$-functions for the four considered input variables (hn1d, hn3d, $z_{\text{pp}}$, $P_{\text{crit}}$) on the training data (AV1) resulted in negligible differences in F1 scores ($\leq 1\%$) compared to the F1-optimized

threshold-based classification. The best performance in terms of F1 and Brier scores was obtained by taking the average probability from $P(\text{AvD})(P_{\text{crit}})$ and $P(\text{AvD})(\text{hn3d})$, which was also confirmed by the validation on the independent dataset from the region of Davos (data set AV3, F1 $= 0.75$). On this data set (AV3), the performance of the $P(\text{AvD})(\text{hn3d})$-model in terms of predicting AvDs was rather low (TPR $= 0.55$). A possible explanation is the more frequent formation of persistent weak layers in the region of Davos due to its relatively dry, inner-alpine snow climate, compared to the mean snow climate

in the Swiss Alps. If weak layers are present within or at the snow surface, avalanches can release with smaller amounts of new snow (e.g. Schweizer et al., 2009; Schneebeli et al., 1998), which was also illustrated by the differences in optimal thresholds for the subsets from the training data (Sect. 4.1). The combination of $P(\text{AvD})(\text{hn3d})$ with $P(\text{AvD})(P_{\text{crit}})$ presents an alternative to using snow-climate-specific thresholds, as the $P_{\text{crit}}$-variable captures the presence of weak layers.

Most of the recently developed snow instability models (Viallon-Galinier et al., 2022; Pérez-Guillén et al., 2022; Hendrick

et al., accepted; Sielenou et al., 2021) are based on statistical methods which account for non-linear, complex relationships between target and explanatory variables. Here, we chose a rather simple approach based on one-dimensional sigmoidal functions which cannot account for interactions between explanatory variables, but allow for a simple interpretation of model output. Nevertheless, it should be noted that $P_{\text{crit}}$ itself is based on the output of a random forest model, which renders the interpretation of $P(\text{AvD})(P_{\text{crit}})$ with respect to the original input parameters of the instability model difficult. For a discussion of the

influence of these input parameters on the direct output of the instability model, the layer-specific probability of instability, $P_{\text{unstable}}$, see Mayer et al. (2022).

For a model to be considered useful, it has to provide more information than can be obtained from basic prior information (Honts and Schweinle, 2009), for instance, when simply assuming the base rate of avalanche days as the constant probability for an avalanche day. Thus, the potential benefits of a threshold-based classification can also be explored using the concept of

information gain (Honts and Schweinle, 2009). Applied to our context, information gain is defined as the difference between the base rate probability of avalanche days and the posterior probability (or the positive predictive value PPV or precision; Honts and Schweinle, 2009). As shown in Table 5 for the avalanche observations in the region of Davos (data set AV3), particularly combining the models $P(\text{AvD})(P_{\text{crit}})$ and $P(\text{AvD})(\text{hn3d})$ provided a clear information gain ($PPV \geq 0.76$) compared to the base rate of avalanche days in this data set ($BR = 0.22$). While the new snow model $P(\text{AvD})(\text{hn3d})$ had a similar PPV, the

combination of the two approaches resulted in a comparably balanced proportion of correctly detected AvDs and nAvDs.

## 5.3 Estimating avalanche size

Avalanche size is classified according to the destructive potential of the avalanche (e.g. EAWS, 2019), which is strongly influenced by the volume and mass of the snow in motion. Thus, avalanche size depends on the failure depth and the extent of the slab which released, the snow entrained in the avalanche path, but also on the terrain itself (e.g. Bartelt et al., 2017). Of

these factors, the one-dimensional snowpack simulations in combination with the instability model only provide information



on the failure depth. To the best of our knowledge, our study nevertheless represents the first attempt to estimate avalanche size from simulated snow stratigraphy.

We estimated avalanche size as a function of various proxies of failure depth (hn1d, hn3d, $z_{pp}$ and $z_{deep}$). The correlation between size and failure depth of observed avalanches was demonstrated here (Fig. 6) and in previous studies (van Herwijnen and Jamieson, 2007). The overall best indicator of the largest avalanche size in terms of Brier scores was obtained with the avalanche size estimator based on the simulated depth of the deepest weak layer, $z_{deep}$. This suggests that information on snow stratigraphy provides important additional information on avalanche size compared to using only indicators related to the amount of new snow. The size estimator based on $z_{deep}$, however, overestimated the occurrence of large avalanches. This might result from the above noted oversimplification of the size estimator with one single input parameter, but also from the quality of the observed avalanche size distributions which represent a single, and often rather small sample (median of two avalanches on avalanche days) from the potential avalanches on a given day. Moreover, there may be a reporting bias towards reporting larger avalanches (Schweizer et al., 2020a; Techel et al., 2020). We cannot account for any of these factors in our analysis.

## 5.4 Comparison with regional avalanche danger levels

The three key factors that characterize the avalanche danger levels are snowpack stability, the frequency distribution of snowpack stability, and avalanche size (Techel et al., 2020; EAWS, 2022). The models developed allow, for the first time, to use a fully data- and model-driven approach to estimate these key factors of regional avalanche danger. We demonstrated that with increasing danger level, the probability for natural avalanches estimated by the avalanche day predictor $P(\text{AvD})(P_{crit})$ also increased for the stations and virtual slopes that matched the elevation and the critical aspects of the respective danger level (Fig. 8, Sect. 4.4.2). Interestingly, the benchmark model $P(\text{AvD})(\text{hn3d})$ separated danger levels 1 (low) and 2 (moderate) from the upper danger levels high (4) and very high (5) even more strictly. This simple model $P(\text{AvD})(\text{hn3d})$ could thus be of particular use for operational forecasting, especially when only meteorological variables and no detailed snow stratigraphy simulations are available. Both avalanche day predictors $P(\text{AvD})(P_{crit})$ and $P(\text{AvD})(\text{hn3d})$ indicated a large bandwidth of conditions for danger level 3 (considerable), suggesting that splitting this danger level into several sub-levels as proposed by Techel et al. (2022) is reasonable. With respect to avalanche size, the predictions of the estimator based on $z_{deep}$ showed a reasonable increase of the probability for large avalanches with increasing danger level, which is consistent with the definition of the danger levels.

The third key factor characterizing regional avalanche danger, the frequency distribution of snowpack stability, was not analyzed in this study. Applying the instability model and the avalanche day predictor to spatially distributed snowpack simulations may yield frequency distributions of snowpack stability with respect to natural release and human triggering, respectively. Spatially distributed simulations of snow stratigraphy can be obtained either with high-resolution output of numerical weather prediction models (e.g. Vionnet et al., 2012; Bellaire and Jamieson, 2013; Horton et al., 2015) or precipitation input scaled according to terrain properties (e.g. Reuter et al., 2016; Richter et al., 2021). While the demonstration of such an approach was out of scope for this study, we here provide an outlook of the combination of the three determinants of regional avalanche danger in comparison to forecast (quality-checked) regional-scale danger levels (Fig. 10) with the data set DL (also used in



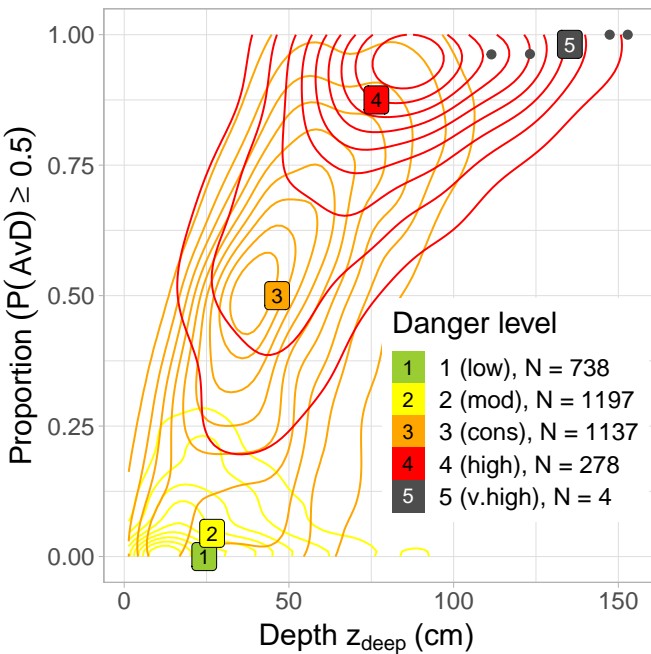

**Figure 10.** Proportion of predictions with $P(\mathrm{AvD})(\mathrm{combi}) \geq 0.5$ and mean depth of deepest weak layer $z_{\mathrm{deep}}$ per day and danger level. Shown are the respective median values (labels). Contour lines indicate the two-dimensional distributions for 2 (moderate), 3 (considerable) and 4 (high). Not shown are density estimates for 1 (low), as cases with proportions $P(\mathrm{AvD})(\mathrm{combi}) \geq 0.5$ of $\geq 0.01$ were rare, and for 5 (very high), as only four data points existed.

Sect. 4.4.2, Figs. 8 and 9). Using posterior knowledge, we aggregated AWSs from regions with the same danger level to estimate a frequency distribution of snowpack stability by the proportion of AWSs indicating natural avalanche occurrence $(P(\mathrm{AvD})(\mathrm{combi}) \geq 0.5)$. Similarly, we calculated the mean approximated failure depth $z_{\mathrm{deep}}$ per day and region with the same danger level. Results suggest that avalanche probability and $z_{\mathrm{deep}}$, the estimator best correlating with avalanche size, increased non-linearly with danger level. The largest spread in conditions can be noted at danger level 3 (considerable), where

the predicted probability for natural avalanches spanned almost the entire range of possible values (see shape of orange density contours in Fig.10).

     Finally, comparing the model-driven predictions of instability related to human-triggered avalanches with other studies exploring the relationship between the determining factors of regional avalanche danger and the danger levels showed similar patterns (Fig. 11a). For instance, studies exploring Rutschblock stability test results, a stability test indicative of human-

triggering of avalanches (Föhn, 1987; Schweizer, 2002), showed that the proportion of test results classified as very poor or poor increased with increasing danger level (Schweizer et al., 2021; Techel et al., 2020, 2022). Similarly, the proportion that at least one human-triggered avalanche was recorded in the area of observation (Schweizer et al., 2021) or the proportion of observations indicating human-triggered whumpfs or shooting cracks (Techel et al., 2022) increased in a similar manner. A





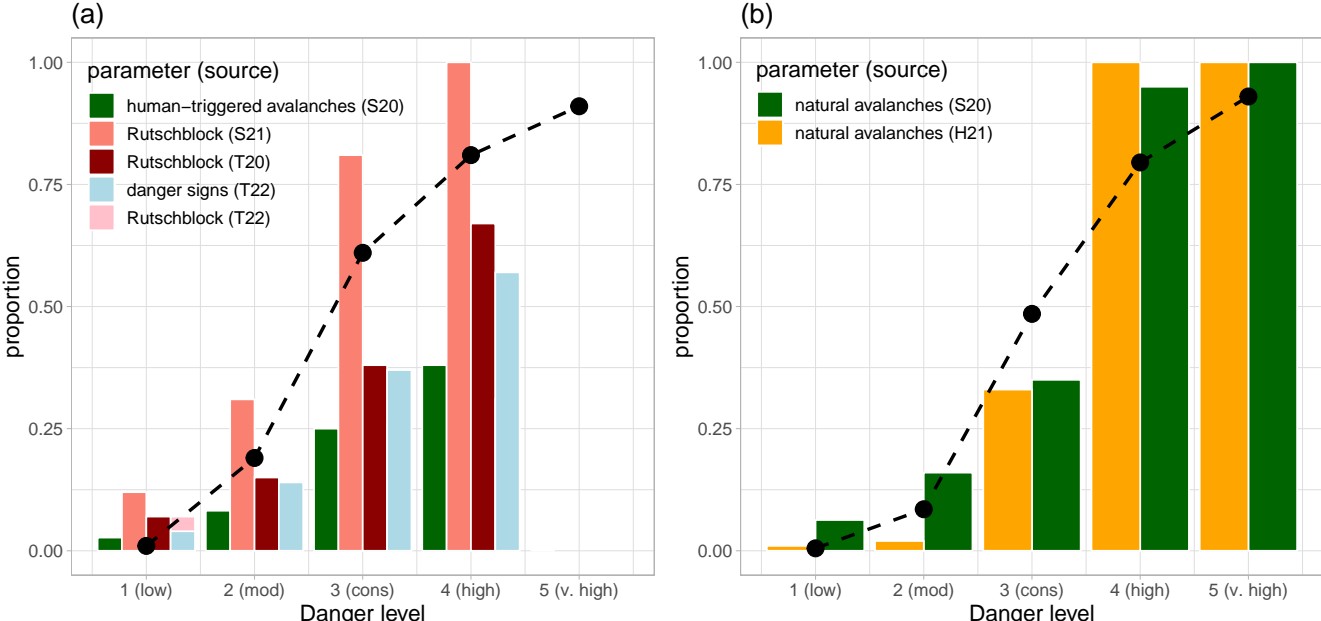

**Figure 11.** Comparison of modelled instability with recent studies analysing observed or forecast indicators of instability with respect to the five danger levels. (a) The proportion of simulated SNOWPACK profiles classified as potentially unstable by the instability model (dashed line) is compared to the proportion of Rutschblock test results, to the proportion of days when a human-triggered avalanche was observed, and to the proportion of observations reporting danger signs. (b) The proportion of profiles that indicated a natural avalanche day based on the combined avalanche predictor $P(\text{AvD})(\text{combi}) \geq 0.5$ is shown in comparison to observed and expected avalanche occurrence. Comparison data sets are abbreviated as follows and refer to the following studies: S20 (Schweizer et al., 2020a), S21 (Schweizer et al., 2021), T20 (Techel et al., 2020), T22 (Techel et al., 2022), H21 (Hutter et al., 2021).

similarly good agreement between model predictions and studies describing the (expected or observed) occurrence of natural
avalanches is visualized in Fig. 11b. For instance, the proportion of $P(\text{AvD})$-predictions, the proportion of days that natural
avalanches were mentioned in the danger description in the Swiss avalanche forecast (Hutter et al., 2021), or that avalanches
were observed (in the region of Davos, Schweizer et al., 2020a) showed low values at 1 (low) and 2 (moderate), and high values
at 4 (high) and 5 (very high). We therefore conclude that fully data- and model-driven predictions describing the probability
of human-triggered avalanches and the occurrence of natural avalanches have a strong correlation with observational data, and
may therefore be suitable to estimate snowpack stability at a regional scale.

## 6 Conclusions

To develop a model describing the probability of natural dry-snow avalanches, we combined the output of a recently developed instability model for one-dimensional SNOWPACK simulations with quality-controlled avalanche observations. The new



avalanche day predictor $P(\mathrm{AvD})(P_{\mathrm{crit}})$ performed well (F1 $= 0.82$), but not better than a benchmark model $P(\mathrm{AvD})(\mathrm{hn3d})$

based on the 3d amount of new snow (F1 $= 0.85$) regarding the classification of avalanche days and non-avalanche days from the training data set of observed avalanches from all over Switzerland (AV1). This suggests that for the occurrence of natural dry-snow avalanches, snow stratigraphy seems to be of secondary importance compared to the amount of new snow. However, model evaluation on an independent data set from the region of Davos (AV3) (Sect. 4.2) and the analysis of specific subsets of the training data showed that accounting for snow stratigraphy is important when prominent persistent weak layers are present

in the snowpack as less new snow is required to cause a decrease in stability. In the classification of avalanche days from the region of Davos, $P(\mathrm{AvD})(P_{\mathrm{crit}})$ outperformed $P(\mathrm{AvD})(\mathrm{hn3d})$ (F1 $= 0.71$ and $0.64$, respectively), and the averaged predictions of both models yielded the overall best performance (F1 $= 0.75$). The performance of this combined model should be evaluated on further independent data sets to investigate its applicability to snow climates that were not represented by the data used in this study.

We also explored whether indicators of avalanche size can be obtained from one-dimensional SNOWPACK simulations. Our avalanche size estimator, developed using observations of avalanche size and failure depth, produced the best results in predicting the largest avalanche size when the depth of the deepest simulated weak layer ($z_{\mathrm{deep}}$) was used as a proxy for failure depth. This demonstrates that including information on snow stratigraphy is critical for estimating avalanche size, compared to relying exclusively on parameters based on the amount of new snow.

The models developed in this study allow for the estimation of two determinants of regional avalanche danger, snow instability and avalanche size. Applied to one-dimensional snowpack simulations driven with data from AWSs or numerical weather prediction models, these models can thus provide valuable support in operational avalanche forecasting.



## Appendix A: Appendix

**Table A1.** Definition of four-parameter sigmoidal functions $f(x)$ used for fitting of $P(\text{AvD})$-functions.

|  | definition |
|---|---|
| logistic | $f_{\log}(x; a, b, c, d) \quad = b + \frac{(c-b)}{1+e^{a(x-d)}}$ |
| modified Gompertz | $f_{\text{gom}}(x; a, b, c, d) \quad = b + (c-b)(1 - e^{-e^{a(x-d)}})$ |
| log-logistic | $f_{\text{llog}}(x; a, b, c, d) \quad = b + \frac{(c-b)}{1+(\frac{x}{d})^a}$ |
| Weibull type 1 | $f_{\text{wei}}(x; a, b, c, d) \quad = b + (c-b)(1 - e^{-(\frac{x}{d})^a})$ |

**Table A2.** Performance statistics for different avalanche day predictors (binary classification). The best-splitting threshold $thr$ is indicated. A case is classified as AvD if the respective value is $\geq thr$, except for sn38 where $\leq thr$.

|  | cross-validation* (median [min - max]) | | | | | all** | | | |
|---|---|---|---|---|---|---|---|---|---|
| parameter | $thr$ | TPR | TNR | PPV | F1 | $thr$ | TPR | TNR | F1 |
| hn1d | 12 [9-17] cm | 0.77 [0.59-0.84] | 0.99 [0.97-1] | 0.84 [0.57-0.98] | 0.73 [0.68-0.83] | 12 cm | 0.74 | 0.99 | 0.80 |
| hn3d | 23 [16-37] cm | 0.81 [0.62-0.94] | 0.98 [0.96-1] | 0.83 [0.56-0.98] | 0.80 [0.66-0.90] | 24 cm | 0.83 | 0.99 | 0.86 |
| $z_{\text{pp}}$ | 32 [22-47] cm | 0.73 [0.51-0.89] | 0.98 [0.96-1] | 0.72 [0.63-0.95] | 0.76 [0.62-0.89] | 31 cm | 0.82 | 0.98 | 0.82 |
| $P_{\text{crit}}$ | 0.81 [0.74-0.85] | 0.79 [0.56-0.93] | 0.99 [0.97-1] | 0.80 [0.69-0.93] | 0.80 [0.62-0.88] | 0.81 | 0.82 | 0.98 | 0.82 |
| sn38 | 1.23 [1.00-1.65] | 0.70 [0.31-0.94] | 0.66 [0.42-0.82] | 0.14 [0.10-0.23] | 0.24 [0.15-0.36] | 1.0 | 0.67 | 0.68 | 0.28 |

*Shown are the median values, and the minimum and maximum values in square brackets.

**The data set *all* was trained and tested on the same data.

**Table A3.** Coefficients (a, b, c, d) of best-fitting function $f(x)$ describing the probability for an AvD, $P(\text{AvD})$, and corresponding Brier score (BS), Brier score on positive events (BS$^+$), and F1 score resulting from classification based on threshold $thr$ with $P(\text{AvD})(thr) = 0.5$. Definition of the functions $f$ are given in Table A1.

| $x$ | $f$ | a | b | c | d | BS | BS$^+$ | F1 | $thr$ |
|---|---|---|---|---|---|---|---|---|---|
| hn1d | modified Gompertz | 0.141354 | -0.117651 | 1.00 | 14.911041 | 0.027 | 0.227 | 0.79 | 13 cm |
| hn3d | log-logistic | -3.295749 | 0.006066 | 1 | 25.997319 | 0.021 | 0.156 | 0.85 | 26 cm |
| $z_{\text{pp}}$ | Weibull type 1 | 1.824612 | 0.004207 | 0.99 | 45.379189 | 0.025 | 0.172 | 0.81 | 37 cm |
| $P_{\text{crit}}$ | modified Gompertz | 11.688441 | 0.004838 | 0.99 | 0.858463 | 0.027 | 0.178 | 0.82 | 0.83 |





**Table A4.** Coefficients $(\beta_0, \beta_1)$ of logistic regression functions $P(S \geq s)(z_{\mathrm{obs}})$ (eq. 4) relating avalanche size $s$ to observed failure depth $z_{\mathrm{obs}}$ (see also Fig. 6b).

| $s$ | $\beta_0$ | $\beta_1$ |
|---|---|---|
| 2 | 0.2916 | 0.0440 |
| 3 | -1.5254 | 0.0242 |
| 4 | -3.6971 | 0.0196 |
| 5 | -6.8279 | 0.0164 |



*Data availability.* The data will be made available at the data repository https://envidat.ch/.

*Author contributions.* SM and FT contributed equally to this study (concept, data curation and analysis, preparation of manuscript; joint first authorship). JS and AvH provided feedback during the analysis and reviewed the manuscript several times.

*Competing interests.* The authors declare that they have no conflict of interest.

*Acknowledgements.* We thank Lukas Graz (ETH Zurich) and Marc Ruesch for valuable advice on the development of the regression models.

*Financial support.* This work was partly funded by the WSL research program Climate Change Impacts on Alpine Mass Movements –
CCAMM (https://ccamm.slf.ch/en/general-overview.html).



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
