# Peer review of "Prediction of natural dry-snow avalanche activity using physics-based snowpack simulations"

_EGUsphere, 2023_

## Author Response (AR1)

Dear Editor, dear reviewer

thank you for your feedback on our replies to the reviewers' comments and to the manuscript, which we greatly appreciate.

Below, we address each of the points raised and describe the changes made in the manuscript. Black text indicates the points raised by you or by the reviewers. The blue text shows our responses to the comments. Line numbers (L) refer to the manuscript including track changes.

**Comments by Editor Pascal Haegeli**

1) In your comments to the reviewers' question, there are several occasions where you clarify a misunderstanding the reviewer might have had, but you do not actually discuss how you will address the issue in the paper to prevent the question from coming up in the future. So instead of just explaining the situation to reviewer, I encourage you to take this opportunity to improve the explanation in the manuscript itself so that future readers do not have the same question. At this point, your responses do not provide any information on whether and how you will be doing this.

We certainly had the intention to include clarifications where needed and you will see below and in the replies to the reviewers how we do so. For example, regarding sn38 (L 157-159, L459-469), the impact of the adapted definition for AvD in the Davos validation data set (L 268-279, L 443-447), or the detailed explanations to Figure 10 (L 411-427).[1]

2) Reviewer 1 highlighted a few circumstances where your terminology and variable names deviate from existing standards (hn3d, depth of precipitation particles). Your response was that these terms are described in the text and that you would prefer to keep them as they are. I encourage you to change your terminology and variables names and ensure that they are in line with existing standards (i.e., ICSSG) and common industry use. In my opinion, these standards were developed to promote consistency and clarity, and it is bad practice to break with them unless there are very strong reasons. As far as I can tell, your reason primarily seems personal preference, which does not seem enough.

We changed the variable names for the height of new snow to be consistent with the ICSSG. All the other variables are non-standard, and we will keep the terminology, for instance for depth. The ICSSG suggests using negative values if thickness is measured from the surface, which is depth. We do not follow this recommendation since it does not make sense in our context.

3) Reviewer 2 pointed out the difference in the avalanche day definition between the training and validation datasets. While your response states that you will improve the description of the definition of the dataset, you did not comment on the implications of this mismatch. I would appreciate if you could elaborate on this in the revised version of your manuscript.

We now explain the potential implications in the revised manuscript (L 268-279, L 443-447). However, it is rather difficult to estimate the influence of this difference in the definition of AvD, especially in a quantitative manner. With the adapted definition of AvD (no consideration of aspect for the two surrounding areas not covered by the avalanche observations in the region of Davos), we
* * *
[1] Given the two-step revision process in NHESS, we are occasionally uncertain how detailed the replies in the Discussion need to be. The present replies will be published as well so that everything is available to the public anyway.

obtain some more AvDs than in the other data sets. This follows from the fact that an east-facing avalanche in the region of Davos will count towards an AvD if there are other avalanches in the surrounding areas with unknown aspect. For the training data set, the ratio AvD/nAvD ranged from 6 to 15%, while for the Davos validation set it was 27%. In other words, the labels for Davos data are less reliable, which may explain why the performance is somewhat poorer. In conclusions, the adapted definition leads to a higher proportion of AvDs, which seems responsible for the lower performance.

4) Reviewer 3 commented on the value of the analysis of the regional avalanche danger level and the importance of Figures 10 and 11 but asked for more details. I personally also think that this is a very interesting part of your study, but it is actually outside of the scope of the present study as you mention on L477. I also noticed that your response to Reviewer 3 did not seem to address their request for additional information.

I am personally not a big fan of these types of "added analyses" in the discussion section for several reasons: a) they present new results that should be described in the result section; b) the associated methods are not described adequately; and c) the potential insight can easily get lost because this aspect of the research is not mentioned in any other parts of the manuscript (abstract, introduction, research objective (L64), etc.). I think you have two options to address this issue: a) remove this comparison/validation with the danger rating from the current manuscript and make it the focus of a separate paper; or b) properly integrate the entire danger rating analysis into the study by including it in the research objective, describing the full approach in the methods, and properly presenting the results (incl. the figures) in the results section. My personal preference is the first option as it creates a better platform for presenting this research to the community in a meaningful way and giving it the attention it deserves.

We prefer to keep the comparison with the danger level in the paper, as we consider these results as important. They underline the overall validity of the model predictions looking from a completely different perspective than validating with avalanche observations. However, we now incorporated these "added analyses" in the main part of the paper, in the Results section (L 411-427). Moreover, we made changes to the conclusions to accommodate that we also relate the model predictions to avalanche danger levels (L 591-594).

Finally, I have several specific questions about your modelling approach:
• It seems to me that the gap check is already fulfilled by the condition for Y = 1 (L183, Equ. 1).

The gap check criterium ensures that we are dealing with a widespread avalanche cycle, and not just with a single large freak avalanche within proximity of the station. For instance, one size 3 avalanche (AAI = 1) observed within the 250 km$^2$ region not only fulfils the conditions for AAI(250 km$^2$) ≥ 0.01 but also for AAI(1000 km$^2$) ≥ 0.04  and AAI(5000 km$^2$)≥ 0.2). In these cases, the gap check requirements AAI(1000 km$^2$) > AAI(250 km$^2$) and AAI(5000 km$^2$) > AAI(1000 km$^2$) ascertain that avalanche activity is not only local but widespread.

• I am curious to know more why you limited yourself to a simple AvD model with only a single predictor (L219)

We aimed at keeping the model as transparent as possible, as $P_{dec}$ already incorporates six snowpack variables. However, the model combining HN and $P_{crit}$ relies on two predictors.

• Why did you choose to estimate multiple logistic models for avalanche size instead of a single ordinal logistic regression model (L233).

We think both approaches would have been fine. The advantage of having multiple logistic models (one for each avalanche size), is that we obtain probabilities for each avalanche size, and not simply

one prediction for the most likely avalanche size. As Schweizer et al. (2020) showed, avalanche activity generally increases across the entire spectrum of sizes with increasing danger level, and not just for the large avalanches.

In the revised manuscript, references are often marked as changed. This is caused by a different .bib-file being used as input compared to the previous version.

References

Schweizer, J., Mitterer, C., Techel, F., Stoffel, A., and Reuter, B.: On the relation between avalanche occurrence and avalanche danger level, The Cryosphere, 14, 737–750, https://doi.org/10.5194/tc-14-737-2020, 2020

**Reply to Reviewer #1**

We thank reviewer #1 for the thorough and detailed review of our manuscript, which we greatly appreciate. In the following, we address each of the points raised. Black text indicates the reviewer's comments. The blue text shows our responses to the comments. Line numbers (L) refer to the revised manuscript including track changes.

**General comments**

This paper investigates the potential to forecast natural dry-snow avalanches with snowpack simulations. Avalanche observation data are used to train and validate models that predict the probability of natural avalanche occurrence, as well as the probability of different avalanche sizes. Comparing to benchmark models that only considered the amount of new snow, they show improved model performance by adding snowpack stratigraphy and stability information, especially in regions prone to persistent weak layers. As a final step, model predictions were validated with regional avalanche danger ratings, which illustrated the potential for this model to support avalanche forecasting.

The study is interesting, thoughtfully designed, and well written (especially considering the complexity of the subject and data sets). The research is relevant to the avalanche research and forecasting community and fits well within the scope of NHESS. I recommend publication after addressing the following comments.

**Specific comments**

My main comment is that it could be clearer how aspect dependent information was applied in the model training and validation. Overall, the data and methods are presented very clearly, but understanding how aspect information was applied required a fair bit of extra effort which I think could easily be improved by providing more details. Some examples:

- Consistent terminology. It would help to explicitly describe values as "aspect-specific" throughout the manuscript when referring to AvD/nAvDs. "aspect-specific" and "slope-specific" are used interchangeably (e.g., lines 255, 336).

  We now consistently use the term "aspect-specific" for the definition of AvD/nAvD (Eq. 1).

- 3.1.1 could provide a clearer description of how avalanche days were counted by aspect in the training data. My interpretation is that for each day of the study period Y is calculated 4 x (number of stations) times and then each value of Y is matched to the explanatory variables for the corresponding virtual slope simulation to build the training dataset. Are the flat field simulations discarded?

  The flat field simulations are not used. We now describe more clearly which data were used; we added an explanation (L 122).

- Line 150 states the height of new snow was independent of aspect, which I assume means these variables were taken from flat field simulations. But if this version of SNOWPACK models snow redistribution, then shouldn't your profiles have different amounts of HN on different aspects and lead to a higher likelihood of natural avalanches on lee aspects?

  Conventionally, the height of new snow is measured in the flat field. Consistent with this definition, the new snow amounts provided by SNOWPACK are therefore for the flat field as

well, regardless of whether it is a simulation in the flat or on a virtual slope. The new snow amounts HN1d and Hn3d considered in our study are therefore indeed independent of aspect. In addition, we also considered the thickness of the layers including precipitation particles as a further parameter in our analysis. For this parameter, which should capture the amount of recently fallen snow including snow transport by wind, we employed the snow redistribution module of SNOWPACK (L151-156).

- Can you comment on the impact of not considering aspect in the AvD definition used in the validation part of the study (i.e., line 253)? The model predicts aspect-specific probabilities, but these are evaluated on whether avalanches were observed anywhere in the region. The relevance of this assumption should be justified.

  We regret that we were not clear; this is a misunderstanding, as aspect-specific information was also used for the AvD definition applied on the validation data. To make it clearer where the AvD definition differs from Eq. (1), Section 3.1.1., we rephrased the paragraph where we describe the definition for the validation data set (L262-279) and briefly discuss the impact of this difference (L443-447).

- Given SNOWPACK's virtual slopes have limited verification studies, it could be interesting to briefly comment on whether this study finds they add value relative to flat field simulations only. Are there any insights or recommendations about virtual slopes to share with future researchers?

  We added a paragraph in which we describe previous work on using slope simulations for the purpose of avalanche forecasting (L556-564).

**Technical comments**

- Introduction: I appreciate how the introduction clearly addressed the limitations of using avalanche observations for model verification. The objective and structure of the paper is also very clear.
- Line 107: Can you state how many avalanches were included in the AV3 dataset (as done for AV1 and AV2)?

  We now provide the number of avalanches in the AV3 dataset in Sect. 2.1.3 (L108).

- Line 120: It may be better to mention wind transport was enabled here rather than line 152. Also, can you describe whether this SNOWPACK setup determines snowfall with precipitation gauges or snow height measurements (since new snow height is an important variable in this study)?

  We now provide more information about snow transport by wind and the SNOWPACK setup (L119-121, L151-159).

- Fig. 1: This figure (and the entire Data section) is organized in a way that makes it very easy to understand the different datasets used in various parts of the study. Thanks.

  We are happy to hear that Fig. 1 is helpful.

- Fig. 2: Can you specify a temporal period for number of avalanche days N(AvD). For example, over a specific study period, seasonal average, or something else?

The AvDs were observed during the three winters 2019/2020 – 2021/2022 as described in Sect. 2.1.1.

- Lines 145-146: Some technical notes on choosing variable names and ICSSG standards. Typically, HN3D would suggest a 3-day observation interval, which is different from the sum of observations made at 1-day intervals (due to settlement). Am I correct that in this case, you are summing HN1D values rather than using SNOWPACK to directly get a height of 3-day snowfall? Also, would it be more accurate to call the precipitation particle variable a "thickness" rather than a "depth"? I would interpret depth as the distance from the deepest PP/DF layer to the surface, but summing thicknesses could be smaller if there are other grain types above (e.g., RG/MF). If that is the case, then ICSSG symbol for thickness is D rather than z. Similarly, the standard symbol for grain type is F. I don't think changing these variables is essential, just something to consider.

  Yes, it is correct that HN3d refers to the sum of HN1d values over three days, which we also stated in the manuscript ("calculated as the sum of three consecutive hn1d-values», l.146). We agree that "thickness of precipitation particle layers" is a more appropriate term than "depth of precipitation particles" and will change the wording throughout the manuscript. Thank you for the comment regarding the ICSSG symbols. However, we prefer to stick to $z_{pp}$ as we think that the definition in line 145 and the illustration given in Figure 3 is sufficient to understand the meaning of the variable.

- Line 148: Can you briefly describe the main inputs of sn38 and how these differ from the inputs to Punstable?

  We added the definition of the natural stability index, which for each snow layer describes the ratio of shear strength to the shear stress exerted by the overlying slab (L157-159)

- Fig. 1: It could be clearer here how many avalanche days were computed (i.e., one per station per day?) and how the aspect and elevation information was used.

  We now describe more clearly how many avalanche days were calculated per station, day and aspect (L200-203).

- Line 181: Variables *st* and *asp* are defined but not used in manuscript.

  We removed these abbreviations.

- 209: Variable *thr* is not defined and appears to only be used in the Appendix.

  We abbreviate the best-splitting threshold using thr. This abbreviation is now introduced (L214).

- Line 208: The subsets are not just based on splits of the AV1 data, but also the snowpack data (i.e., critical grain type). Perhaps it's better to say "we split the training data…".

  We changed the wording according to your recommendation and specify the term "training data" in the previous paragraph (L 200-201, 213, 218-219).

- Line 220: Why was sn38 only used in the binary model and not the continuous model? It would help to list the x variables in the beginning of this subsection (since the variables used in the binary model aren't described either).

Due to the limited discriminatory power of sn38 in the binary approach, this variable was not considered further in the subsequent development of continuous models. We included a sentence to clarify this (L316-318). We discuss findings regarding sn38 on L 459-469.

- Line 228 and 243: The idea behind the BS+ metrics could be a bit clearer. Throughout the manuscript they are described as "minority class", "positive observed outcomes", "positive events", "when condition is fulfilled". I recommend a sentence to explain why these subsets are relevant in the methods and then choosing consistent terms that are more descriptive (e.g., only days with observed avalanches) to use throughout the manuscript.

Thank you for this recommendation. We now describe the idea behind the BS+ score more clearly and use more descriptive wording in the revised version of the manuscript (L238-241, 256-258).

- Line 252: Why were aspect and elevation neglected when determining AvD here?

Unfortunately, the region of Davos where we have aspect- and elevation-information does not cover the whole area (i.e. 5000 km$^2$) that we need to consider in the definition of an AvD. As the avalanche observations outside the region of Davos sometimes lacked information on aspect and elevation, we had to adjust the gap-check requirements. While aspect and elevation were considered for the avalanches from the region of Davos, for the two surrounding regions, this was not the case. A day was labeled as an aspect-specific AvD, if the AAI in the region of Davos was larger than 0.01 for the respective aspect and within an elevation band of ±250 m around the AWS WFJ, and if at least one natural dry-snow avalanche was observed within each of the two surrounding regions (1000 km$^2$ and 5000 km$^2$) regardless of aspect and elevation. The definition of an AvD was thus slightly adapted compared to Eq. (1) due to the lack of consistent information on aspect and elevation of the observed avalanches within the two larger surrounding regions. We now explain this more clearly in the revised version of the manuscript (262-279, 443-447).

- Line 284: According to Table A2 the median is 12 cm not 13 cm.

Thank you for pointing out this error. 12 cm would have been correct, as shown in Figure 4 and Table A2.

Figure 4 is not cited anywhere in the manuscript.

You might have overseen that Figure 4 is mentioned in line 286/7, which was probably due to the line break.

- Fig. 5: The 2020 season seems to stand out as anomalous in these figures, was there something unique about that season, such as the prevalence or absence of persistent weak layers?

The 2020 season was a rather mild winter, with comparably few avalanche periods (Trachsel et al., 2020). This is also shown in Table 3. However, we have no conclusive information on whether persistent weak layers were of less concern during this winter compared to the two other seasons. Table 3 does not show evidence for either assumption.

- Line 316: Should this be dataset AV2 instead of AV3?

Yes, thank you for pointing out this error, this should be data set AV2.

- 7: The HN3D model is presented before the Pcrit model in Fig 5, while it's the other order in Figs 7 and 8. Consistent ordering would be ideal.

  We changed the order of the plots in Figure 5, with the $P_{crit}$-model being shown before the HN3d-model.

- Lines 371-380: This paragraph is a little confusing because not all the values discussed are shown in Fig. 9b, which appears to be due to whether the Pdeep < 0.77 cases are included or not. Which case is more relevant to present here? The text could be clearer about which case is being discussed and which case is shown in the figure.

  We restructured this paragraph to make it easier to understand (L 399-403, 408-410).

- Line 395: I am particularly interested in how the extreme cases of widespread versus no activity impact the results at danger level 3. Since the models are fit to these extreme cases, they do not capture the "natural avalanches possible in certain areas" conditions experienced at danger level 3. However, it seems the models produce a desirable result with a wide range of P(AvD) observed at considerable danger in Fig. 8, which speaks to the range of conditions and uncertainty about natural avalanches experienced at considerable danger.

  To develop the models for the prediction of natural avalanche activity, we put a lot of effort into the quality of the target variable data (AvD vs. nAvD). By cross-checking the activity in smaller regions (250 km$^2$) with the activity in larger regions (1000 and 5000 km$^2$) and retaining only the data points at the two ends of the stability spectrum (widespread vs. no activity), we aimed at reducing the error contained in the visual observations of local observers. To overcome this binary approach, we fitted continuous functions describing the probability for an AvD, P(AvD), forcing the models to be close to 0 at nAvD data points and close to 1 at AvD data points. With the very broad definition for danger level 3 (considerable), some of the validation data points are closer to the conditions of AvD data points in terms of $P_{crit}$ or HN3d, and others may be closer to the conditions of the nAvD training data points. This explains the wide range of P(AvD)-values obtained at danger level 3 (considerable). Techel et al. (2022) also demonstrated the capability of $P_{crit}$ to capture the differences in stability regarding sub-levels describing variations within danger levels.

- Line 446: While not published in a peer-reviewed journal, Bellaire & Jamieson (2013) estimated avalanche size from simulated profiles in their 2013 ISSW paper (Fig. 2 is similar to your Fig. 6a).

  Thank you for pointing this out, we were not aware of this study and now provide reference to this work in the revised version of the manuscript (L 513, 522-526).

- Fig. 10: This caption could provide higher-level description of what is shown, as it is difficult to understand without also reading the text. Also, what exactly do the contour lines show? Steps in density distributions?

  Yes, the contour lines indicate steps in density distributions. More specifically, the respective outermost contour line represents the (0, 0.1] percentile interval, and the innermost contour line the (0.9, 1.0] interval. We now describe that in detail in the caption to Figure 10.

- Fig. 11: Why are the pink bars for Rutschblock (T2) only shown for low danger and not others?

  Thank you for pointing this out. This was in fact an error, which we corrected.

- Conclusion: Given the stated objective of the paper was to "*investigate whether the instability model developed by Mayer et al. (2022) applied to one dimensional SNOWPACK simulations can be used to predict natural dry-snow avalanches*", I think this question can be more directly answered somewhere in the conclusions to summarize what was learned and how others could apply the model.

  To answer this question more directly, we added a sentence at the beginning of the Conclusion section.

**References**

Bellaire, Sascha, and Bruce Jamieson. "On estimating avalanche danger from simulated snow profiles." In Proceedings of the International Snow Science Workshop, Grenoble–Chamonix Mont-Blanc, pp. 7-11. 2013. https://arc.lib.montana.edu/snow-science/item.php?id=1740

Hafner, E. D., Techel, F., Leinss, S., and Bühler, Y.: Mapping avalanches with satellites – evaluation of performance and completeness, The Cryosphere, 15, 983–1004, https://doi.org/10.5194/tc-15-983-2021, 2021.

Techel, F., Mayer, S., Pérez-Guillén, C., Schmudlach, G., and Winkler, K.: On the correlation between a sub-level qualifier refining the danger level with observations and models relating to the contributing factors of avalanche danger, Natural Hazards and Earth System Sciences, pp. 1911–1930, https://doi.org/10.5194/nhess-22-1911-2022, 2022.

Trachsel, J.; Zweifel, B.; Techel, F.; Marty, C.; Stucki, T., 2020: Schnee und Lawinen in den Schweizer Alpen. Hydrologisches Jahr 2019/20. WSL Ber. 101: 76 S.

**Reply to Reviewer #2**

We thank reviewer #2 for the thorough and detailed review of our manuscript, which we greatly appreciate. In the following, we address each of the points raised. Black text indicates the reviewer's comments. The blue text shows our responses to the comments.

The authors use simulated snowprofiles from SNOWPACK around AWS as both avalanche day predictor and avalanche size predictor for natural dry avalanches. They validate these indicators using a long-term dataset of re-analyzed danger levels and a short-term avalanche observation dataset. They further show how their predictors improve on simple avalanche day indicators by adding information about stratigraphy and stability.

The study is very well written with good and clear figures and tables. Additional information is found in the appendix. I find the study interesting for a large part of the avalanche community both scientific and practical.

My main critique is that used datasets and methods were developed in other studies. It is very time-consuming to read all these associated studies to gain a better insight into the methods. This is especially true for the binary avalanche day classification.

The discussion does a good job at listing a handful of limitations of this study, however, falls short in explaining what impact these limitations have. I am pointing towards some of these limitations in my comments below.

20: please reconsider this sentence. You write that erroneous forecasts may cause costs as false alarms may lead to economic loss. Isn't that the same thing?

Forecasts may be erroneous in two ways, false alarms and misses. Both may cause costs, which we explain with this statement.

23: I think accurate forecasting of natural avalanches in space will never be possible. You write about forecasting the location of avalanches also in the first sentence in the abstract which is a bit misleading.

We doubt that accurate forecasting of natural avalanches in space will never be possible, as physics-based models and computational power will likely further improve in the future. Moreover, accurately forecasting natural avalanches in space does not necessarily refer to the scale of a slope but can also refer to regional-scale activity. We removed the word "location" in the abstract.

Figure 1: great figure that is very helpful in understanding which datasets serve which purposes.

Thank you, we are glad that this figure is helpful.

97: it seems like you are referring to Table 3 before you refer to Table 2.

We removed the first reference to Table 3, as we want it to remain close to the Results section. Table 2 is now referred before referring to Table 3.

130: what is the rational behind selecting the deepest WL as Pcrit? If I understand your method correctly, the DH layer at around 50 cm in Figure 3 should be Pcrit? Thank you for clarifying this!

This is probably a misunderstanding. We defined $P_{crit}$ as the maximum value of $P_{unstable}$ over the respective profile. The values of $P_{unstable}$ of the deepest weak layer indicated by the instability model was termed $P_{deep}$. Figure 3 should clarify this. The sentence "In case of ties, we selected the layer deepest in the snowpack" means that if the maximum of $P_{unstable}$ is not unique, i.e. the maximum of $P_{unstable}$ is assigned to more than one layer across the profile, then the values for $z_{crit}$ and $gt_{crit}$ (depth and grain type of the weak layer) are taken from the deepest of those layers. Only very few cases had non-unique values of $P_{crit.}$

148: I am not familiar with sn38. Could you explain very briefly where and how it is used?

We added the definition of the natural stability index sn38, which for each snow layer describes the ratio of shear strength to the shear stress exerted by the overlying slab on a 38° slope (L157-159) and discuss our findings regarding sn38 on L 459-469.

150: What is the rational behind assuming that new snow height is not aspect dependent? You describe in the sentence below that there is a snow redistribution routine in SNOWPACK.

New snow depths used in avalanche forecasting are usually given by representative flat field measurements in line with meteorological precipitation measurements. In accordance with this, the height of new snow provided by SNOWPACK is always the value for the flat field, even for the slope simulations. The new snow amounts HN1d and HN3d considered in our study are therefore indeed independent of aspect. We added an explanation (L 151-156).

In addition, we also considered the thickness of the layers of precipitation particles as a further parameter in our analysis. For this parameter, which should capture the amount of recently fallen including snow transport by wind, since we employed the snow redistribution module of SNOWPACK.

Section 3.1.1: I had to obviously read the Hendrick et al. (accepted) paper to better grasp your definition of avalanche days. I do not think that it is particularly favorable to the readability and comprehensibility of this paper, that one must read up on the methods in another paper. I, however, do think that the algorithm is clever.

We refer to Hendrick et al. (2023) as this is the publication where we first developed this definition of an avalanche day definition. However, we included all essential information in our manuscript making it not necessary to read the paper by Hendrick et al. (2023), which was published in the meantime.

There are a couple of things that I was wondering about, that do not necessarily have to be answered in a revised manuscript:

- How do the different gap check requirements compare to the size of the forecasting regions (I know that you have dynamic regions) as well as to the typical size one of your observers can cover to do avalanche observations?

  This comparison is one of the few points, which we did not repeat from the description given in Hendrick et al. (2023): 250 $km^2$ is approximately the size of the spatial units used in the forecast in Switzerland, while 5000 $km^2$ is close to the average size of the seven snow-climatological regions in the Alps.

- We often say that avalanches are rare events and given that you have had a median of two avalanches per aspect and elevation in an area of 250 km2 around an AWS, I wonder if this is true?

In our data set, avalanche days were indeed comparably rare (we had ten times more nAvD than AvD). We do not know the number of avalanche release areas in the aspect and elevation band surrounding a station, but presumably a median of two avalanches represents only a small fraction of the potential release areas, and thus of potential avalanches, in this area. Hence, avalanches may still be considered a comparably rare event, even on avalanche days.

209: I might have missed it, but what is "thr"? Threshold?

We abbreviate the best-splitting threshold for binary classification using thr. This abbreviation is now introduced (L 214).

Section 3.1.2: this section was very hard for me to comprehend and only after reading the results from 4.2 onwards, it became more clear what you were doing.

We revisited this section and tried to make it clearer.

Section 3.3: Is it a problem that there is a mismatch between the number of observed avalanches per aspect and elevation within 250 km2 of an AWS in the training dataset (N=2) and the number of observed avalanches regardless of aspect and elevation within 1000 and 5000 km2 which is a minimum of 1? In my mind you are getting more avalanche days due to a less stringent threshold in your validation dataset than in your training dataset.

It is correct that the avalanche day definition is slightly less stringent compared to the definition used for training and testing the P(AvD)-models. To make it clearer where the avalanche day definition differs from Section 3.1.1., we adjusted the paragraph, where we describe the definition for the validation data set ( L262-279) and discuss the results (L 443-447).

255: Do I understand you correctly that you are using SNOWPACK simulations forced by Weissfluhjoch data for the entire validation dataset?

It is correct that the for the validation part concerning the avalanche activity data from the region of Davos (data set AV3), SNOWPACK simulations were forced with Weissfluhjoch data. We added this information (L266-267).

260: What is the rationale behind removing simulated snow depth < 30 cm?

We removed data points that had a simulated snow depth of less than 30 cm, as avalanches are very unlikely to release with such shallow snowpacks. We thereby mainly reduced the number of data points labeled as non-avalanche days. We consider the days with sufficiently high snow depths as more relevant.

263: Do you mean that avalanche days were in general associated with new snow, both 24 and 72 hours?

Yes, this is correct, we therefore provide both median values (L286-287).

284: 12 or 13 cm?

Thank you for pointing out this error: It should read 12 cm, as is shown in Figure 4 and in Table A2.

297: Interesting observation about persistent weak layers needing less new snow for natural triggering. I thought that there was not much difference between the strength of persistent and non-persistent weak layers during and immediately after snow fall. However, non-persistent forms sinter quicker than persistent ones (Alec's lab study from 2013). And I believe that Ben Reuter and others showed in 2018 that non-persistent forms were initially as weak as persistent forms, however, gaining in strength quicker.

In this paragraph, we compared the optimal threshold for the thickness of precipitation particle layers $z_{pp}$ to differentiate between AvD and nAvDs for different subsets. We found that the threshold for $z_{pp}$ was higher for the case when the critical weak layer determined by the instability model consisted of precipitation particles compared to the case when it consisted of persistent grain types. From this, however, we cannot directly deduce that persistent weak layers need less overloading than non-persistent weak layers, because the variable $z_{pp}$ does not describe the depth of the weak layer (i.e. the non-persistent weak layer could be anywhere within the precipitation particle layers contributing to $z_{pp}$).

309: What is the physical explanation for taking the mean of both models?

There is no clear physical explanation for taking the mean of both models. The two models may be inaccurate in different situations, and by averaging the models, one model's strength can balance out the other model's weakness to some extent.

317: A median failure depth of 30 cm for size 1 avalanches is surprisingly high in my mind. Where were you surprised about that result? I must confess that I am positively surprised that simulated weak layer depth is such a good predictor of avalanche size. I thought that discerning avalanche size is much more complex.

A median failure layer depth of 30 cm may seem high, considering all size 1 avalanches. Presumably, there is a bias towards reporting "larger" size 1 in our data set. Interestingly, Bellaire and Jamieson (2013), in an ISSW proceedings paper pointed out by Reviewer 1, also observed a median failure layer depth of 30 cm for size 1 avalanches. We now refer to this publication in the revised manuscript (L 513, 522-526).

Figure 7: median values are not always readable.

We increased the font size for these median values.

Figure 8: the numbers indicating respective portions above and below threshold are not always readable.

We increased the font size for these proportions.

395: With the target variable including either a lot or no avalanche activity, what would you expect your results to be if medium avalanche activity days were accounted for? How does the time stamp of 12:00 LT for your model simulations influence the results? Do you foresee some problems with regards to when observers record avalanche activity during the day? I am also convinced that "medium avalanche activity" might characterize many avalanche days with considerable avalanche danger (ref Fig 8).

Our approach to train the model using a binary target variable (widespread vs. no avalanche activity) was driven by the demand for high-quality labeling of the training data. We would also like to emphasize that the criterion defining the AvD does not require "a lot" of avalanche activity within

the 250 km$^2$ surrounding of the AWS, but solely an AAI of at least 0.01, which corresponds to a single small avalanche. With the criterion for increasing avalanche activity within the larger surroundings (1000 km$^2$, 5000 km$^2$), we, however, ensured that activity was widespread and thereby presumably reduced the number of data points where single observers allocated observed avalanches to a wrong date.

Defining a third label with "medium avalanche activity" is difficult. Which avalanche activity index would this label refer to? And which P(AvD)-value would we assign to medium avalanche activity? With our approach, we decided to go the other way around, conduct the training with only two labels, but evaluate on a data set covering a much wider spread of activity. The evaluation on the danger level set then provided plausible results, regarding the wide range of P(AvD)-values on days with considerable avalanche danger (level 3), which contains days with "medium avalanche activity" as you mentioned.

The timestamp of 12:00 LT for the model simulations influences the results in so far, that for example the critical new snow amount may be under-/overestimated if avalanches occurred several hours after/before this timestamp. To overcome this uncertainty more accurate data with exact avalanche release times would be necessary.

469: …or get rid of the avalanche danger levels altogether (just my personal opinion and somewhat confirmed by Figure 10)

500: pretty interesting!

**References**

Bellaire, S. and Jamieson, B. "On estimating avalanche danger from simulated snow profiles."
In Proceedings of the International Snow Science Workshop, Grenoble–Chamonix Mont-Blanc, pp. 7-11. 2013. https://arc.lib.montana.edu/snow-science/item.php?id=1740

**Reply to Reviewer #3**

We thank Christoph Mitterer for the thorough and detailed review of our manuscript, which we greatly appreciate. In the following, we address each of the points raised. Black text indicates the reviewers' comments. The blue text shows our responses to the comments.

**Summary**
The authors introduce and investigate the predictive skills of two new models: an avalanche day predictor and an avalanche size estimator. Both are based on snow cover model results and observed avalanche activity and especially designed and valid for natural dry-snow avalanche events. The models are trained and tested on individual data sets. The training data consists of two large data sets of avalanche observations and snowpack simulations using the 1-D physics based model SNOWPACK. The validation data sets include avalanche observations, avalanche danger level assessments and snow cover simulation results.
The model for the avalanche day predictor focuses on a RandomForest (RF) model based on derivates of SNOWPACK variables presented by the authors team very recently (Mayer et al., 2022) and is trained using a 3-years data set of avalanche observations data covering the entire Swiss Alps. The estimator model is trained with observed data only, but includes a very large data set of avalanche observations covering 30 years for the Swiss Alps.
To test the performance of both models, the authors test their novel approaches against a very common benchmark model (Height of the three-day sum of new snow; hn3d) and validate their findings against a fully independent data set of avalanche observations. Large parts of the interpretation and discussion is done by comparing the results to a 21-years data set of regional avalanche danger levels assessment.

Results show good predictive performance results for characterising days with natural dry-snow avalanche activity; especially when natural dry-snow avalanche activity was driven by shallow snowpacks consisting predominantly by persistent weak layers. Compared to the simple benchmark model, performance is very similar and increases slightly when the new approach is combined with the benchmark model. The results for the avalanche size estimator when tested again the independent data set are also encouraging. Finally, both approaches are compared to the regional avalanche danger levels.

**Evaluation**
The approaches are not fully novel but connect skilfully recent advances with a large data set that represents the currently available golden standard within the avalanche research community. Research objectives are very clear and concise; methods are well designed, but also how the authors addressed the objectives are mostly well described and easy to grasp.
Language is concise and the manuscript is well written. Major findings are very relevant to the avalanche research and forecasting community. The content fits very nicely into NHESS. There are a few parts of the discussion and interpretation of results that need revision. Having addressed the below stated comments, I recommend publication.
**General comments**
I have the following general comments:

- In the Abstract (Line 1) and Introduction (Lines 19-21) you give the reader the feeling that you would like to tackle both, very local (path scale) and regional avalanche forecasting. When reading the full manuscript, it becomes obvious that you address regional avalanche forecasting (e.g., Section 3.1.1 or the fact that you compare and discuss results to the

regional avalanche danger level). Please be more specific in that case and drop the connection to the local avalanche forecasting.

We now make it clearer in the Abstract and Introduction that we are aiming at predicting regional-scale avalanche activity (L1, 65).

- The authors compare their avalanche day predictor model to the conventional natural stability index on a 38° steep slope (sn38) which was one of the few indices developed within SNOWPACK to better assess natural avalanche activity (Lehning et al., 2004). The new model outperforms the sn38 (Figure 4), but the authors do not really discuss why this is the case. They state that "The poor performance of sn38 is in line with other studies (Schweizer et al., 2006; Jamieson et al., 2007)." To my knowledge the first study tackles sk38 only, thus skier-triggered scenarios and not spontaneous avalanche activity. The second study compares the natural stability index (sn) based on measurements to natural avalanche activity in the surroundings of the study plot – which is a significant different approach to the one presented (modelled vs. measured). In fact – to my knowledge the only qualitative investigation on the performance of sn38 is given in Lehning et al. (2004). There the authors show reasonable results.
So, it remains difficult to set the presented low performance skills into context. Therefore, it would be very interesting and valuable to tackle in more detail the question, why sn38 has such a low performance compared to the avalanche day predictor model. Both, the conventional and the novel approach, are heavily parameterised by snow density and almost rely on the same concepts: the most important variable for the instability model by Mayer et al. (2022), the viscous deformation rate, shares the identical input parameter as the natural stability index, namely natural shear strength – which in turn is parametrised using snow density. 4 out of the 5 most important features building the RF model rely on snow density. It would be very beneficial for the community if the authors could e.g. use the Weissfluhjoch data set to shed some light into this topic. I know that this represents large efforts, but I believe it would give even more impact to presented results.

We included the natural stability index sn38 to evaluate the new approach based on the output of the instability model in direct comparison with such a benchmark model. The index sn38 is defined as the ratio of weak layer shear strength divided by the shear stress due to the overlying slab. This criterion seems well-suited to model natural avalanche activity from a physical point of view. However, the parametrization of this simple criterion within SNOWPACK has some weaknesses: While the shear stress can be simply calculated from the load and thus only inherits the errors from estimating precipitation mass based on measured snow depths, shear strength is a rather complex microstructural parameter. The current SNOWPACK parametrization of shear strength is based on density and grain type (Jamieson and Johnston, 2001), which may not be sufficient to capture the influence of microstructure as also pointed out by Richter (2020). In particular, the evolution of the SNOWPACK shear strength over time only depends on density if grain type does not change.
In their comparison of modelled sn38 values with forecasted danger levels, Lehning et al. (2004) pointed out a tendency of the natural stability index to indicate maximum weaknesses close to the ground which they attributed to an underestimation of the increase in shear strength due to its simplified parametrization within SNOWPACK. An underestimation of the increase in shear strength could also be an explanation for the low precision of the sn38, which led to many false alarms in our study. We thus agree that your question may warrant a

detailed analysis of sn38 but this would be out of the scope of the present study, as field data including observed snow strength would be necessary.

Finally, you are correct when stating that Schweizer et al. (2006) analyzed the skier stability index sk38 rather than the natural stability index. We included a reference to this paper as they stated, "the natural stability index is a poor predictor for spontaneous releases". But as they did not prove this statement within their study, we will delete this reference in the revised version of the manuscript. Instead, we now refer to Reuter et al. (2022) who also demonstrated a rather poor performance of the sn38 and showed that using time derivatives of this index has a higher predictive power. This is in line with Jamieson et al. (2007) who analyzed sn38 based on field measurements and also concluded that critical values of stability indices are less useful than their trends.

We discuss the findings regarding sn38 on L459-469.

- Interconnected to the comment above: How and why is the instability model suited to predict natural avalanche activity, even though it is heavily trained on data that mostly represents skier-triggered avalanche activity?

  It is correct that the instability model predicts the probability of (potential) instability, related to human-triggered avalanches. However, the input features of the instability model describing the weak layer (e.g. grain size) and the overlying slab (e.g. ratio of the mean slab density and the mean slab grain size) are also related to the release of natural avalanches as mentioned in line 55 f. While not mentioned in Mayer et al. (2022), we would like to point out that often natural avalanches were reported in the vicinity of the "unstable" profiles used for the training of the instability model. And lastly, we think that our results are in line with our understanding of snowpack stability: according to Mayer et al. (2022), profiles are classified as unstable if pcrit > 0.77, and – in this study – we find natural avalanches are likely if pcrit > 0.82. Thus, the latter is a subset of the profiles classified as unstable from a human-triggering perspective.

- The discussion regarding the comparison to the regional avalanche danger levels is nice but needs in a few points a much broader approach: The statement that danger level 3-Considerable needs sub-levels could also be reversed in the fact that the Swiss forecaster need to train themselves more in order to transfer the overlapping parts into the neighbouring classes instead of increasing the level of discretization. Can you comment on that please. Figures 10-11 are very important but touched very shortly. I would appreciate more details here.

  We agree that our results are based on how the avalanche danger levels are assigned in the Swiss avalanche bulletin. However, the broad range of what is considered danger level 3-Considerable, shown in Figure 10, seems not to be a Swiss bias but rather inherent to the broad definition of the danger levels. This broad range, from a rather low proportion to a rather high proportion of locations where natural avalanches may initiate, is also mirrored in the EAWS matrix (EAWS, 2023), a look-up table assisting forecasters with danger level assessments in Europe. The EAWS matrix suggests for level 3 (considerable) combinations ranging from "Many locations with very poor stability (which corresponds to natural avalanches) exist. Avalanches can reach size 2." and "(Nearly) no locations with very poor stability but some locations with poor stability (human triggered avalanches are typical for

this class) exist." The first definition has a tendency towards level 4 (high), which is reflected both in the figure and in the EAWS matrix. This broad range of 3 (considerable) also mirrors what is described in Swiss avalanche forecasts (Hutter et al., 2022), again in line with the EAWS matrix, and what can be seen, when comparing it to actual observations, stratified by sub-levels as used in Switzerland (as in Techel et al., 2022).

**Specific and technical comments**

- 1 (Lines 48-49): Why don't you address all danger levels here? In fact, at danger level 3-Considerable the definition mentions: In certain situations, some large, and in isolated cases very large natural avalanches are possible.

  On purpose, we only addressed the respective danger levels for which natural avalanches are generally expected (4 (high) and 5 (very high)) or not expected (1 (low) and 2 (moderate)). At 3-Considerable, the range is much wider, including situations when avalanches are primarily triggered by additional load as a skier and situations when natural avalanches are typical (e.g., EAWS, 2023).

- 1.1. Line 96: Counter for Table Numbering is not in sequential order. You mention Table 3 before you mention Table 2 in the text.

  Thank you for pointing this out. We removed the first reference to Table 3, as we want the table to remain close to the Results section. Thus, Table 2 is now referred to before Table 3.

- 2 It would be very helpful to introduce a new habit when using SNOWPACK simulations, namely placing the INI-Files of the model runs into the Appendix.

  We will publish the INI-Files of SNOWPACK runs together with the data used to build the new models.

- 1.1 Definition of avalanche days and non-avalanche days (Lines 197-198): Does that mean that your training data set has no AvD due to a size 4 avalanche?

  No, the values and ranges indicated represent the median and the interquartile range (IQR). The latter includes only the 25 to 75%-range, thus, larger values, as for instance size 4 avalanches as the largest avalanche, exist.

- 3.1.2. Avalanche size estimator: Could you please specify in a little more detail, why you have chosen exactly this approach?

  As our instability model is based on assessing the stability of a simulated profile, we focused on fracture depth as an indicator of avalanche size, since we thought it to be feasible to estimate fracture depth from a simulated profile.

- 4.2 Line 376: Figure 9e does not exist.

  Thank you for pointing this out. It should read Figure 9b.

**References**

EAWS, 2023: EAWS Matrix. https://www.avalanches.org/wp-content/uploads/2022/12/EAWS-Matrix-EN.pdf

Hutter, V., Techel, F., and Purves, R. S.: How is avalanche danger described in textual descriptions in avalanche forecasts in Switzerland? Consistency between forecasters and avalanche danger, Natural Hazards and Earth System Sciences, 21, 3879–3897, 2021.

Jamieson, J. B., and C. D. Johnston, Evaluation of the shear frame test for weak snowpack layers, Annals of Glaciology, 32, 59–68, 2001.

Jamieson, J. B., Zeidler, A., and Brown, C.: Explanation and limitations of study plot stability indices for forecasting dry snow slab avalanches in surrounding terrain, Cold Regions Science and Technology, 50, 23–34, 2007.

Lehning, M., Fierz, C., Brown, R. L., and Jamieson, J. B.: Modeling instability for the snow cover model SNOWPACK, Annals of Glaciology, 38, 331–338, 2004.

Mayer, S., van Herwijnen, A., Techel, F., and Schweizer, J.: A random forest model to assess snow instability from simulated snow stratigraphy, The Cryosphere, 16, 4593–4615, 2022.

Reuter, B., Viallon-Galinier, L., Horton, S., van Herwijnen, A., Mayer, S., Hagenmuller, P., & Morin, S.: Characterizing snow instability with avalanche problem types derived from snow cover simulations. Cold Regions Science and Technology, 194, 103462, 2022.

Richter, Bettina. Improving numerical avalanche forecasting with spatial snow cover modeling. Dissertation. ETH Zurich, 2020.

Schweizer, J., Bellaire, S., Fierz, C., Lehning, M., and Pielmeier, C.: Evaluating and improving the stability predictions of the snow cover model SNOWPACK, Cold Regions Science and Technology, 46, 52–59, 2006.

Techel, F., Mayer, S., Pérez-Guillén, C., Schmudlach, G., and Winkler, K.: On the correlation between a sub-level qualifier refining the danger level with observations and models relating to the contributing factors of avalanche danger, Natural Hazards and Earth System Sciences, pp. 1911–1930, 2022.